



# Temporal evolution of Red Sea temperatures based on in-situ observations (1958-2017)

Miguel Agulles[1], Gabriel Jordà[1,2], Burt Jones[3], Susana Agustí[3], Carlos M. Duarte[3,4]

[1]Instituto Mediterráneo de Estudios Avanzados (UIB-CSIC), Esporles, Spain
[2]Centre Oceanogràfic de Balears. Instituto Español de Oceanografía. Palma, Spain
[3]Red Sea Research Centre (RSRC), King Abdullah University of Science and Technology, Thuwal 23955, Saudi Arabia
[4]Computational Bioscience Research Center (CBRC), King Abdullah University of Science and Technology, Thuwal 23955, Saudi Arabia

**Abstract**. The Red Sea holds one of the most diverse marine ecosystems in the world, although fragile and vulnerable to ocean warming. Several studies have analysed the spatiotemporal evolution of the temperature in the Red Sea using satellite data, thus focusing only on the surface layer and covering the last ~30 years. To better understand the long-term variability and trends of the temperature in the whole water column, we produce a 3D gridded temperature product (TEMPERSEA) for the period 1958-2017, based on a large number of in situ observations, covering the Red Sea and the Gulf of Aden. After a specific quality control, a mapping algorithm based on optimal interpolation has been applied to homogenize the data. Also, an estimate of the uncertainties of the product has been generated. The calibration of the algorithm and the uncertainty computation has been done through sensitivity experiments based on synthetic data from a realistic numerical simulation.

TEMPERSEA has been compared to satellite observations of sea surface temperature for the period 1981-2017, showing good agreement specially in those periods with a reasonable number of observations were available. Also, very good agreement has been found between air temperatures and reconstructed sea temperatures in the upper 100 m for the whole period 1958-2017 enhancing the confidence on the quality of the product.

The product has been used to characterize the spatio-temporal variability of the temperature field in the Red Sea and the Gulf of Aden at different time scales (seasonal, interannual and multidecadal). Clear differences have been found between the two regions suggesting that the Red Sea variability is mainly driven by air-sea interactions, while in the Gulf of Aden, the lateral advection of water also plays a relevant role. Regarding long term evolution, our results show only positive trends above 40 m depth, with maximum trends of 0.045 + 0.016 ºC decade-1 at 15 m, and the largest negative trends at 125 m (-0.072 + 0.011 ºC decade-1). Multidecadal variations have a strong impact on the trend computation, and restricting them to the last 30-40 years of data can bias high the trend estimates.



## 1 Introduction

The Red Sea is a narrow basin, meridionally elongated (2250 Km), lying between the African and the Asian continental shelves, and extending from 12.5 °N to 30 °N with an average width of 220 Km (Figure 1). It is a semi-enclosed basin connected to the Indian Ocean through the Bab-al-Mandeb Strait at the south and to the Mediterranean Sea through the Suez Canal at the north. The bathymetry is highly irregular along the basin, with a relatively shallow mean depth (524 m; (Patzert, 1974), but with

maximum recorded depths of almost 3.000 m. At its northern end, it bifurcates into two gulfs, the Gulf of Suez on the West with an average depth of 40 m and the Gulf of Aqaba on the East with depths exceeding 1.800 m (Neumann and McGill, 1961).

The transport through the Suez Canal, which connects the Mediterranean Sea with the Gulf of Suez and the Red Sea, is relatively small, and therefore, the only significant connection between the Red Sea and

the global ocean is the Strait of Bab-al-Mandeb (Sofianos et al., 2015). Due to its arid setting, the Red Sea experiences one of the largest evaporation rates in the world, which in combination with its semi-enclosed nature leads to high salinities across the whole basin (Sofianos et al., 2015). The hydrodynamic characteristics are strongly influenced by the wind forcing with different seasonality. The seasonal winds blow south-eastwards in the northern part of the basin through the whole year, but in the southern region,

the winds reverse form north-westerly in summer to south-easterly in winter under the influence of the two distinct phases of the Arabian monsoon, (Patzert, 1974; Sofianos, 2015).

The Red Sea holds one of the most diverse marine ecosystems in the world, although fragile and vulnerable to ocean warming (Thorne et al., 2010). Water temperature plays a key role in ecosystems evolution, which are usually adapted to the environmental thermal range. Marine species respond to

ocean warming by shifting their distribution poleward and advancing their phenology (Poloczanska et al., 2016). While parts of the ocean may be warming gradually, others may experience rapid fluctuations, inducing more significant impacts on biodiversity. Impacts of warming are likely to be greatest in semi-enclosed seas, which tend to support warming rates higher than the global ones (Lima and Wethey, 2012), as documented for the Red Sea (Chaidez et al., 2017).

Several recent studies have analysed the spatiotemporal evolution of the temperature in the Red Sea using satellite data from AVHRR (Advanced Very High-Resolution Radiometer), thus focusing only on the surface layer and covering from early 1980's onwards. Those studies have identified a warming trend with values ranging from 0.17 ℃ decade-1 to 0.45 ℃ decade-1  across the basin for the period 1982-2015 (Chaidez et al., 2017). Also, sea surface temperature exhibits a strong interannual variability (Eladawy et

al., 2017) which is mainly driven by the air temperature (Raitsos et al., 2011).  However, these studies are limited to ~30 years due to the observational period of remote data. Also, although the evolution of surface conditions is very relevant, the temperature variability in the whole water column has effects on marine biota (Bongaerts et al., 2010), so products based on depth-resolving in situ observations better reflect the thermal regime across the ecosystem than sea surface trends alone.

Global hydrographic products like EN4 (Good et al., 2013) or ISHII (Ishii and Kimoto, 2009) that interpolate in-situ observations to create a monthly 3D product for the last decades are available. However, those products have low spatial resolution (~1°) and the quality controls applied are not region specific, which cast doubts on their accuracy in the narrow Red Sea. In order to overcome the limitations


satellite products and global hydrographic products have, and to be able to characterize the spatiotemporal variability of the 3D temperature field to inform research on the thermal ecology and variability of Red Sea ecosystems, a dedicated regional observational product is required.

Here we produce a gridded temperature product for the period 1958-2017 at monthly resolution as a resource to describe the evolution of the Red Sea temperature during the last six decades and underpin research on the impacts of ocean warming across the Red Sea. The product covers the Red Sea and the

Gulf of Aden with a spatial resolution of 0.25°x 0.25°. This product is based on the assimilation and reanalysis of a large number of in situ observations collected in the region. After a specific quality control, a mapping algorithm has been applied to homogenize the data. Also, an estimate of the accuracy of the product has been generated to accurately define the uncertainties of the product. We then use the product to characterize the seasonal, interannual and multidecadal variability of the 3D temperature field

in the Red Sea and the Gulf of Aden.

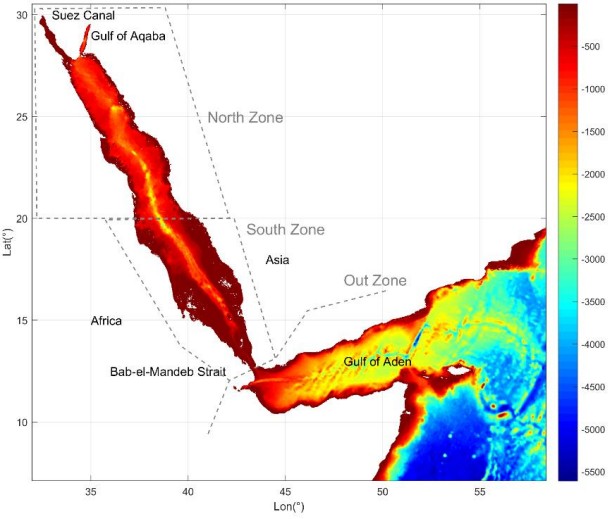

**Figure 1: Domain and bathymetry of the region included in the TEMPERSEA product. The three zones used in the presentation of results (North, South and Outer) are identified by grey lines. (data source: https://www.gebco.net/data_and_products/gridded_bathymetry_data).**

## 2 Data and Methods

### 2.1 In situ data

In situ temperature observations were obtained from two databases. The first one is CORA (Cabanes et al., 2013), a delayed mode product designed to feed global reanalyses. CORA covers the global ocean from 1950 to 2016 and integrates quality controlled historical profiles from several data collections

(Argo, GOSUD, OceanSITES and World Ocean Database). The details of this database can be found in http://www.coriolis.eu.org/Science2/Global-Ocean/CORA and it  is freely delivered by the Copernicus




Marine              Service              (http://marine.copernicus.eu/services-portfolio/access-to-products/?option=com_csw&view=details&product_id=INSITU_GLO_TS_REP_OBSERVATIONS_013_001_b).

The second source of data is the database collected by King Abdullah University of Science and Technology (KAUST), from 2010 to 2018. It includes all the data collected by KAUST in the Red Sea through different platforms (floats, ships, gliders, Argo). The data has been quality controlled with specific criteria for the Red Sea and will be used here as the reference dataset, (Karnauskas and Jones, 2018).

**2.2 Satellite data**

Two sources of remote sensed sea surface temperature (SST) data are used. The first dataset is obtained from the National Ocean and Atmosphere Agency (NOAA) and is based on AVHRR (Advanced Very High-Resolution Radiometer) over the period 1981 -2017. These data have a spatial resolution of 0.25°x0.25°, and can be obtained at monthly temporal resolution from the National Center for
Environmental Information (NCDC-NOAA, ftp://ftp.ncdc.noaa.gov/pub/data/).

The second source of data is OSTIA, a global product generated by UK Met Office (Roberts-Jones et al., 2012). OSTIA merges in situ data from the ICOADS dataset with satellite data from infra-red radiometers over the period of 1985 to 2007. The dataset has a spatial resolution of 0.25°x0.25° and monthly temporal resolution,                              (http://marine.copernicus.eu/services-portfolio/access-to-
products/?option=com_csw&view=details&product_id=SST_GLO_SST_L4_REP_OBSERVATIONS_010_011). To complete the data from 2007 to 2018 another L4 OSTIA product is used, (Bell et al., 2000). Both OSTIA products are merged after a cross validation is performed. The data is available at (http://marine.copernicus.eu/services-portfolio/access-to-products/?option=com_csw&view=details&product_id=SST_GLO_SST_L4_NRT_OBSERVATIONS_0
10_001).The main difference between NOAA and OSTIA products is that the later uses the in situ data to correct the satellite data, (Roberts-Jones et al., 2012).

**2.3 Model data**

The outputs from a realistic numerical model are used to perform synthetic observational experiments that help to calibrate the mapping algorithm. The model chosen has been the GLORYS.S2V4 global
reanalysis. It is performed with NEMOv3.1 ocean model with a horizontal resolution of 0.25° and 75 vertical z-levels. It is forced by ERA-Interim atmospheric fields (Dee et al., 2011) for the period 1993 to 2015. GLORYS assimilates along track satellite observations of sea level anomaly, sea ice concentration, SST and in situ profiles of temperature and salinity from CORA data base. More details can be found in (Garric and Parent, 2018) and the data is available at (http://marine.copernicus.eu/services-
portfolio/access-to-products/?option=com_csw&view=details&product_id=GLOBAL_REANALYSIS_PHY_001_025).




### 2.4 Data quality control

Prior to the generation of the gridded product it is important to be sure that individual profiles are reliable. In situ profiles in CORA have been quality controlled using an objective procedure and a visual quality

control (Cabanes et al., 2013). However, the objective quality control process was originally tuned for the global ocean, therefore requiring an additional review of the profiles inside the region of interest by a visual quality control.

First, we have reviewed all the profiles to remove spikes, out layers and density inversions. In a second step, we have checked the consistency between the CORA profiles and the profiles collected by KAUST,

which are considered to be more reliable as they have been thoroughly analysed by the KAUST data centre with specific criteria adapted to the region. That assessment has been performed separating the Red Sea profiles (North and South regions in Figure 1) from the profiles obtained south of the Bab-al-Mandeb strait, (Outer region, see Figure 1).

All the profiles in both regions are shown in (Figure 2), along with the 1st and 99th quantiles of the

KAUST profiles (only inside the Red Sea, black lines). It can be seen that two different regimes appear inside the Red Sea and are clearly identifiable by the temperatures below 500 m. Those profiles with temperatures colder than 20ºC below 500 m show a behaviour which is typical of the outside region, while such pronounced cooling with depth is absent from the KAUST profiles. Thus, those profiles are probably misplaced inside the Red Sea, which coldest temperatures at depth exceeds 20 ºC, were rejected.

For the rest of the profiles, those lying outside a range defined by three times the standard deviation (blue lines in Figure 2) are also rejected.

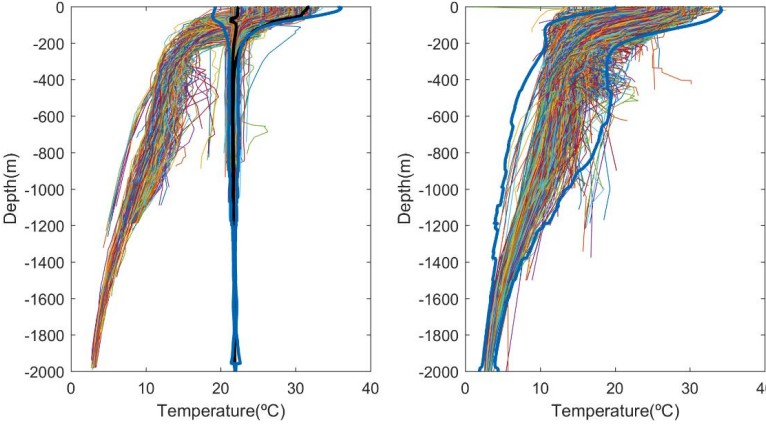

**Figure 2: CORA profiles inside the Red Sea (left) and in the outer region (right). The 1st and 99th quantiles of**
**the KAUST profiles are shown in black. The range defined by 3 times the STD of the CORA profiles is shown**
**in blue.**

After applying the quality control, 11191 profiles are kept inside the Red Sea (82 % of initial profiles) and 30522 are kept in the outer zone (88 % of initial profiles). The number of observations per year and per zone is shown in Figure 3. For the outer zone, there is a large number of observations reaching more than



1500 profiles in some years except during the period 1990-2000 in which the number of observations decreased. Regarding the Red Sea, the number of profiles per year in both zones is usually around 200, although in some periods there is a noticeable lack of data (e.g. during the 70s and between 2004 and 2010). In 2001, in the North zone, there is a peak of observations due to an intensive campaign carried out during the summer of that year.

Considering the number of observations per month (Figure 3), we can see that it remained almost constant through the year in the Southern zone. In contrast, the Northern region is more density sampled in summer, reaching up to more than 1000 observations in July, with roughly 500 observations on average per month during the rest of the year. Regarding the outer zone, the number of observations per month are between 2000 and 3000, with more samples obtained during the first half of the year.


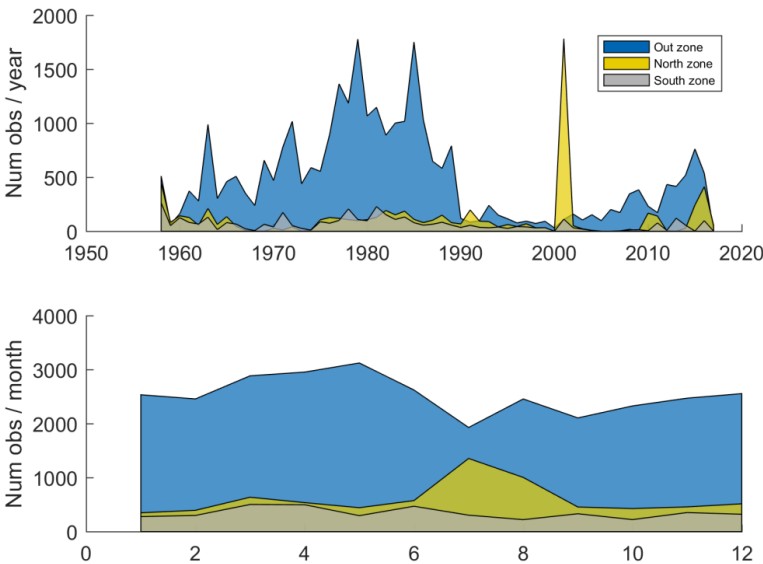

**Figure 3: Number of observations per year (top). Number of observations per month (bottom). North zone in yellow, South zone in grey and Outer zone in blue.**

### 2.5 Mapping Algorithm

In situ observations provide a basis for many oceanographic and meteorological applications. However, the number of observations is limited in space and time and statistical methods must be often applied to homogenize the dataset to be fitted for climate studies and/or model validation (Larsen et al., 2007). We used a classical optimal interpolation algorithm (henceforward OI) to generate 3D gridded monthly temperature maps from individual in situ profiles (henceforward called TEMPERSEA product).

OI is an algorithm that estimates the optimal value of the field as a linear combination of available observations and a background (i.e. first guess) field, with weights determined from the statistics of




observational and background errors. The weights are obtained minimizing the variance of the analysis error (e.g. Jordà & Gomis, 2010). Assuming we have N observations to be mapped into M grid points, the analysed field at a given position r can be written in matrix form as:

$$\hat{U}(r) = BK + S^T * D^{-1} * d \tag{1}$$


Where $BK$ is a M-vector with the background field, S is a NxM matrix containing the covariances of the field between the observation and grid locations, D is a NxN matrix containing the covariances between, and d is the N-vector of observed anomalies with respect to the background field:

$$d = y_{obs} - BK(r_{obs}) \tag{2}$$

The observations are not perfect, and assuming that observational errors are not correlated with the true field, the covariance matrix D can be split into the sum of two matrices;

$$D = (B + R) \tag{3}$$

where the elements of $B$ describe the covariance of the true field between pairs of observation points $(B_{ij} = \overline{\hat{U}(r_i)\hat{U}(r_j)})$ and R contains the observational error covariances $(R_{ij} = \overline{\varepsilon_i\varepsilon_j})$. In our case we

assume observational errors are decorrelated, so R becomes a diagonal matrix with observational error variances in the diagonal. To sum up, the value of the analysis field at point r is given by

$$\hat{U}(r) = BK + S * (B + R)^{-1} * d \tag{4}$$

For convenience, covariance matrices can be transformed to correlation matrices dividing by the field variance $\tilde{\sigma}^2$ This implies that the diagonal elements in $R$ are now defined as $\gamma_2 = \frac{\varepsilon^2}{\sigma^2}$, the noise-to-signal

ratio. The correlations of the field between different locations and times is modelled using a Gaussian function for the spatial component and an exponential por the temporal component:

$$\rho = e^{\frac{-d_{ij}^2}{2*L^2}} * e^{\frac{t_{ij}^2}{T}} \tag{5}$$

Where $d_{ij}$ is the distance between points $i, j$ , and $t_{ij}$ is the time lag. $L$ is the spatial correlation length scale, and $T$ is the time correlation scale.

The parameters, $L, T$ and $\gamma$ , have been determined from sensitivity experiments using synthetic data. In particular, GLORYS fields are considered as the "truth". Temperature profiles are extracted from GLORYS outputs at the same time and location than the actual profiles were obtained. Then, the mapping algorithm is applied to those synthetic profiles and the outputs from the analysis are compared to the original GLORYS fields. Thus, we can estimate the optimal value for L and γ parameters that minimizes



the error of the mapping algorithm provided the characteristics of the observational network and the field variability. The parameter $T$ has been estimated computing the autocorrelation time scale from the GLORYS fields. The analysis is performed over a grid with a spatial resolution of 0.25°x0.25° and 23 vertical depth levels unevenly distributed (Figure 4).

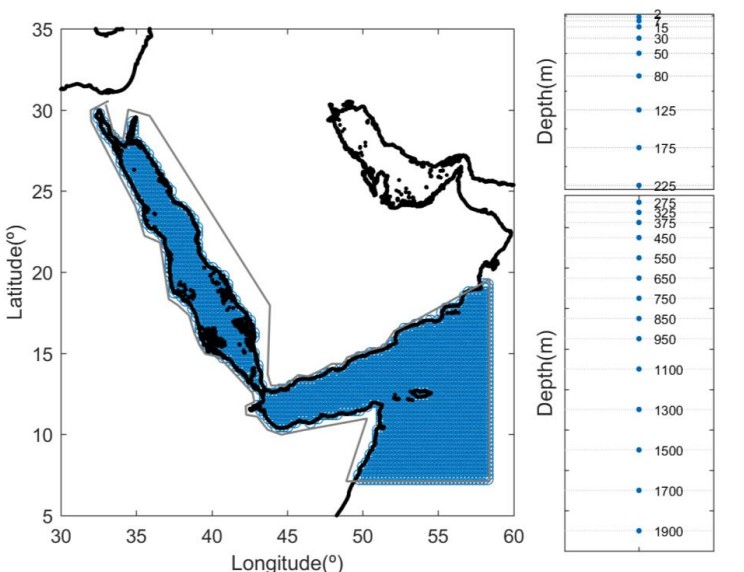

**Figure 4: (Left) Analysis grid used in the generation of TEMPERSEA product. (Right) Depth levels.**

The background (BK) used in TEMPERSEA is a 12-month climatology. This climatology is computed merging all available observations for each month and then applying the OI algorithm to them. However, the number of profiles is often large and the inversion of the $D$ matrix in (1) can be ill-conditioned when profiles are too close (i.e. at a distance much lower than the correlation length scale). Thus, before the OI

algorithm is applied, a data thinning is performed using a K-means algorithm. This clustering technique divides the whole set observations into a predefined number of clusters (Camus et al., 2011). In this case, each cluster represents the mean value of all the observations close to a centroid location. An example is presented in Figure 5. for the month of January. Once we have defined the reduced set of observations grouped per months, the OI algorithm (1) is applied using $L_{back} = 150$ Km and $\Upsilon_{back} = 0,1$. No time

correlation is considered for the computation of the climatology. Those parameters have been obtained from sensitivity experiments as explained before, which also have shown that the data thinning does not degrade the quality of the background field.

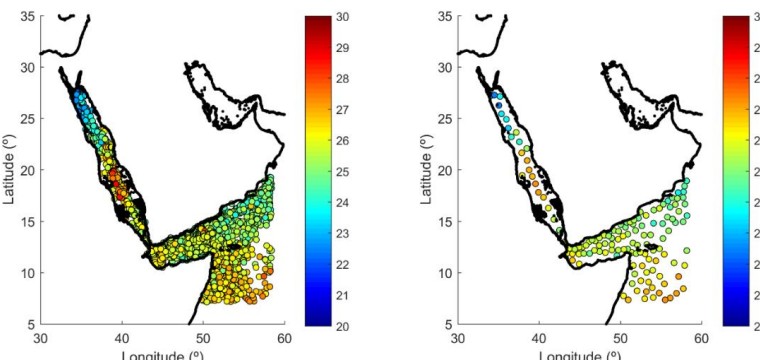

**Figure 5: (Left) distribution of all the available observations for January (n=2705). (Right) distribution of observations after applying the K-means algorithm (n=135).**

Once the climatology is computed, the analysis is performed on the anomalies with respect to it. Thus, the total temperature is computed as the combination of the background and the analysed anomaly field (see (4)). For months or locations lacking observations the analysis will tend to the deliver background field (i.e. second term in the right-hand side of equation (4) is zero). The parameters used for the analysis are L= 200 km, T= 2 months and γ= 0,1. An example of the results of the analysed temperature anomalies for two consecutive months is shown in Figure 6.

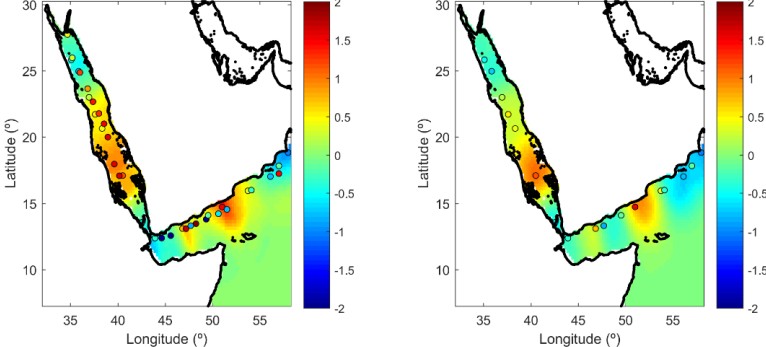

**Figure 6: Analysed temperature anomaly field for October 1958 (left) and November 1958 (right). The dots represent the location and value of the observations used in the analysis of each month.**

### 2.6 Product error

One of the advantages of the OI formulation is that it also provides an estimate of the error covariances (or correlations) associated to the analysis. The MxM analysis error covariance matrix $\widehat{\Sigma}$ is given by

$$\widehat{\Sigma} = G - (S^T(B+R)^{-1}S) * \widetilde{\sigma}^2 \tag{6}$$

Where the entries of the $MxM$ matrix $G$ are the correlations of the background error between pairs of analysis points, $S$, $B$ and $R$ have been defined above and $\widetilde{\sigma}^2$ is the variance of the field. The latter is estimated from the outputs of GLORYS model. We are particularly interested in the diagonal terms of $\widehat{\Sigma}$, which give the analysis error variance ($\widehat{\varepsilon}^2$) at each of the $M$ analysis points.

The formal estimate of the analysis error variance given by (6) depends on the number and distribution of observations as well as on the parameters chosen, but not on the observations themselves. Therefore, it is useful to have a first-order estimate of the accuracy of the formal error estimates. To do so we perform a test using synthetic data from GLORYS outputs. That is, we extract pseudo-observations from GLORYS temperature fields, apply the mapping algorithm as defined above and compare the outputs with the original model fields to obtain the "true" errors. Figure 7a shows the time evolution of the RMS difference between the GLORYS outputs and the background ($\sqrt{\overline{\sigma^2}}$) averaged per vertical levels, where the std of the errors in the background field was used as an estimate of the error in the temperature field. This is an important quantity as it defines the baseline error that our product has in places/times when no observations are available. Error estimates are largest at 125 m depth, with a clear seasonality in the upper layers. Below 300 m the background errors decrease well below 1ºC, except in some periods at 1000 m in which GLORYS data show strong deep anomalies. Concerning the spatial distribution of the background errors (Figure 8a), values average 0.57ºC at 7 m, with higher values along the Arabian coasts and in the Gulf of Aden, where background errors reach 1ºC. At 125 m the averaged background error is 1.12 ºC, with minimum values in the central Red Sea (0.5ºC) and maximum values in the Gulf of Aden (~1.5ºC), where interannual variability is more important and the climatological background is less representative of the temperature field. Figure 7b shows the time evolution of the RMS error of the analysed temperature field is presented. Although the main features seen in the background errors are present, it is clear to that using OI improves the estimate of the temperature field compared with the use of climatology, with a reduction rate ranging from 1.3 to 1.6 (i.e. errors reduced between 30% and 60%). The RMS error maps at 7 m show the averaged value to be reduced to 0.44 ºC, and in most areas the reduction rate is larger than 1.5. At 125 m the RMS error is larger again in the Gulf of Aden but the reduction rate ranges between 1.2 and 1.5. Finally, the formal error estimates are slightly lower (about 20% lower) than the "true" error (Figure 7c and Figure 8 right). However, it is able to capture the seasonality, the maximum at 125 m, and the higher values around 1200 m. The error decreased between 1975 and 1990, due to the higher number of observations distributed in space and time during that period (Figure 7), as also reflected in the "true" error. The formal error replicates the same spatial structure as the "true" error does, both at the surface layer and at 125 m. The magnitude of the formal error is slightly lower than the "true" error (basin averages are 0.31ºC and 0.44ºC, respectively, at the surface, and 0.71ºC and 0.88ºC, at 125 m).

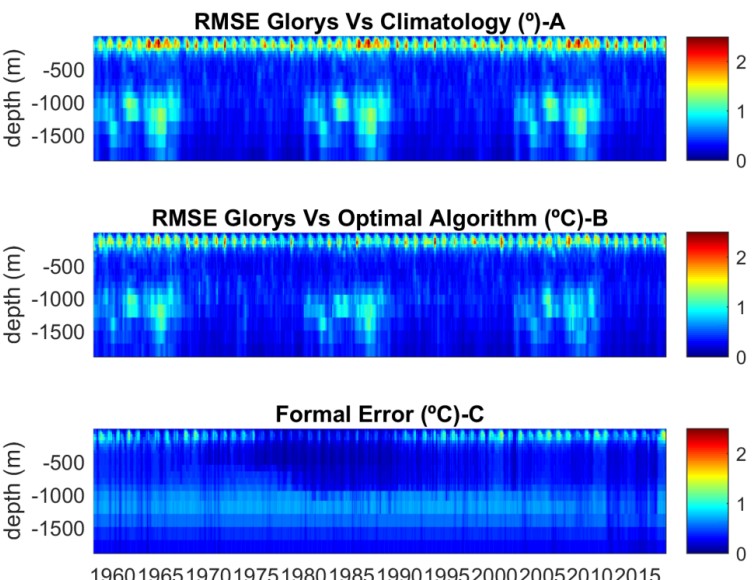

**Figure 7: (a) Standard deviation ($\sqrt{\overline{\sigma^2}}$) of the background errors (in ºC). (b) RMSE (in ºC) of the analysis fields**
**obtained using synthetic data from GLORYS. (c) Formal error ($\sqrt{\widehat{\varepsilon^2}}$, in ºC).**

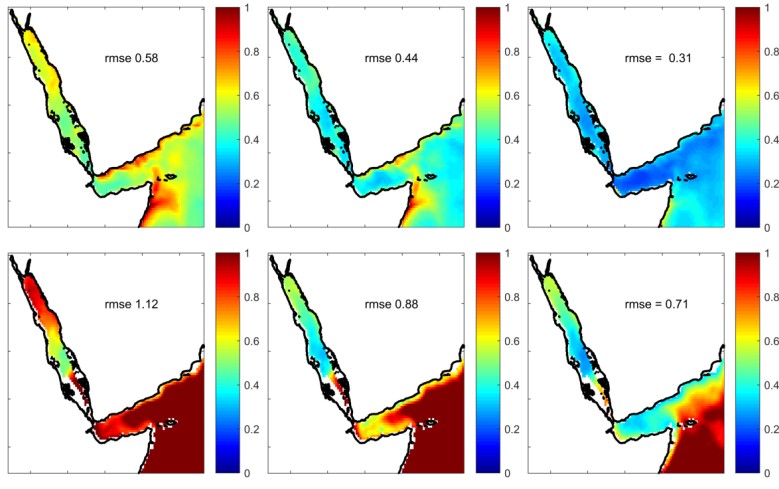

**Figure 8: Horizontal distribution of the RMSE (in ºC) of the background (left) and OI (centre), and formal**
**error (right). The results are shown for 7 m depth (top) and 125 m depth (bottom).**

Computing the formal error using (6) allows us to derive the formal estimate of the errors when
computing regional averages. The formal estimate of the regional average can be computed as:




$$\varepsilon_{av}^2 = \frac{1}{M^2} \sum_{ij} e_i e_j \qquad (8)$$

Where $\varepsilon_{av}^2$ represents the error of the average, $M$ is the length of analysis points and $\sum_{ij} e_i e_j$ is the sum of the error covariances between all the pairs of points included in the averaging. Figure 9 shows the time evolution of the formal estimate of the average temperature in the north zone (in grey) with the "true" error obtained from synthetic observations (in blue). Obviously, the formal error cannot capture the actual error at each month (i.e. is a statistical approximation), but it can be seen that it fits the std of the "true" errors. Also, it is able to identify the periods in which the errors decrease (between 1975-2000) due to the larger density of observations. Therefore, the formal estimate seems to be a reasonable estimate of the analysis error.

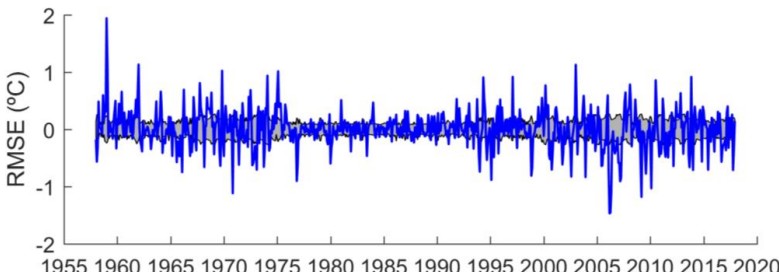

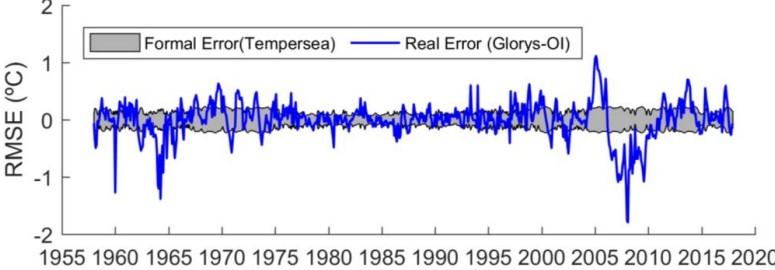

**Figure 9: Comparison of the formal error of the temperature average in the North area with the "true" error when the algorithm is applied to synthetic data. Results shown for (a) 7m depth and (b) 125 m depth.**

## 3. Results

### 3.1 Comparison with satellite results

We compared the first level fields with satellite data from AVHRR and OSTIA as an independent evaluation of the TEMPERSEA product. The monthly variability of regionally averaged temperature anomalies from TEMPERSEA shows good agreement with the satellite estimates (Figure 10). Monthly variations of ~1ºC are captured by all the products as well as variations at lower frequencies. During the periods with few observations the analysis anomalies tend to zero, so the discrepancies with the satellite products increase. The correlation with AVHHR ranges between 0.42 and 0.61 and between 0.39 and 0.51 with OSTIA (Table 1). Discarding the periods with few in situ observations (i.e. with formal error > 0.15ºC) the monthly correlations reach 0.67 and 0.61, respectively. Regarding the RMSE the values range between 0.43 and 0.48 ºC for AVHRR and 0.38 and 0.49 ºC for OSTIA. It must be noted that the SST



value of TEMPERSEA corresponds to the first level of the product, 2 meters. This level represents the
mean value of the profile temperature from the surface to 4 meters of depth. In contrast, the satellite
products take the value of the temperature on the first mm of the water column, and consequently the
variability of the satellite data is larger than TEMPERSEA. Finally, both satellite estimates, although
highly correlated (0.86-0.91) show important discrepancies, with RMS differences of 0.21-0.24ºC (Fig.
10, Table 1).

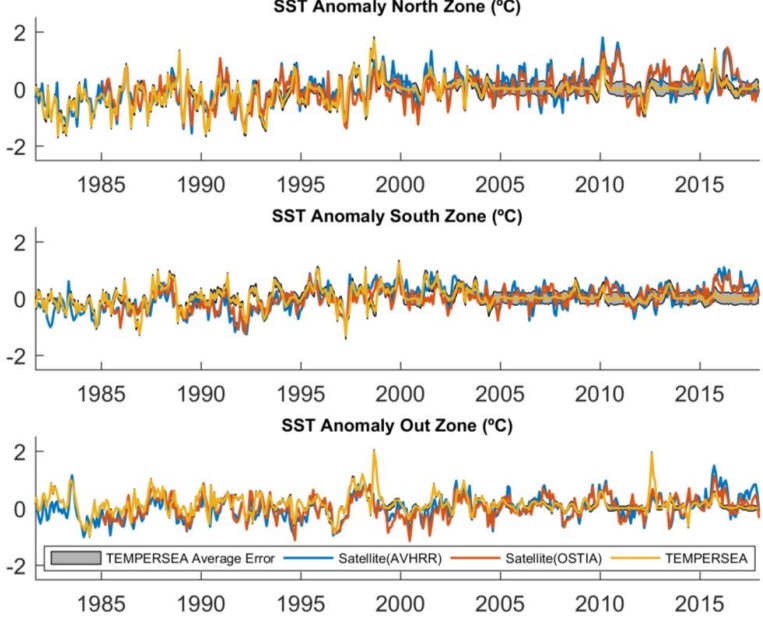

**Figure 10: Monthly anomalies of regionally averaged SST in the three zones, north (top), south (middle) and
outer (bottom). The three datasets are shown for the common period: Tempersea (yellow line), AVHRR (blue
line), OSTIA (red line). The grey patch represents the uncertainties estimated for the TEMPERSEA product.**

|  | North zone | | South zone | | Outer zone | |
|---|---|---|---|---|---|---|
|  | Correlation | RMSE (ºC) | Correlation | RMSE (ºC) | Correlation | RMSE (ºC) |
| TEMPERSEA - SAT(AVHRR) | 0.61(0.67) | 0.48(0.47) | 0.42(0.48) | 0.43(0.48) | 0.47(0.58) | 0.43(0.44) |
| TEMPERSEA - SAT(OSTIA) | 0.51(0.61) | 0.49(0.50) | 0.39(0.47) | 0.38(0.44) | 0.43(0.47) | 0.41(0.43) |
| SAT(AVHRR) - SAT (OSTIA) | 0.91 | 0.24 | 0.86 | 0.23 | 0.87 | 0.21 |

**Table 1: Statistics of the comparison of regionally averaged SST monthly anomalies between TEMPERSEA,
AVHRR and OSTIA. In brackets the values when only periods with enough in-situ observations (i.e. formal
error <0.15ºC) are considered.**

**3.2 Monthly Climatology**

We used TEMPERSEA to characterize the thermal regime of the Red Sea. The averaged field at the
surface is characterized by temperatures ranging from 25.5°C in the northern part of the Red Sea to 29ºC





in the southern part, with a strong gradient at around 20°N (Figure 11a). In the outer region SST's are lower ranging from 26.5°C in the Indian Ocean to 28°C in the Gulf of Aden. Temperatures > 23.5°C are found until a depth of 125 m inside the Red Sea, while only above 50 m in the Gulf of Aden (Figure 11b, c). Below those depths there is a contrasting difference between the two regions, with temperatures in the
335 Red Sea being relatively stable (~22-24°C) through the water column, while temperature decrease almost linearly with depth in the Gulf of Aden, reach 5°C at 1500 m depth, due to the oceanic influence (Sofianos et al, 2015).

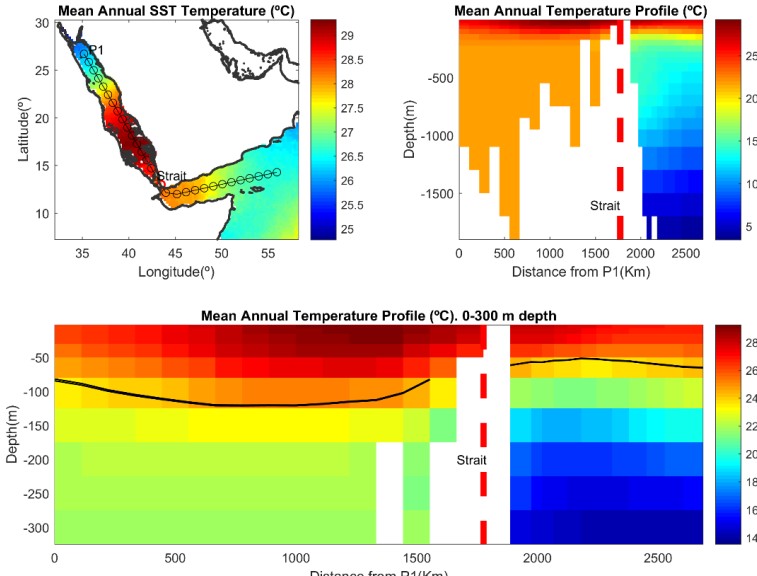

**Figure 11: Average temperature from TEMPERSEA computed for the period 1958-2017. (a) Averaged SST.**
**Dots indicate the location of the vertical section shown in the following figures. (b) Vertical section (c) zoom of the vertical section for the upper 300 m. The black line indicates the isotherm of 23.5 °C.**

The seasonal thermal evolution in the Red Sea, characterized as the anomaly of the monthly climatology relative to the annual mean, is characterized by negative anomalies in surface temperatures relative to the annual mean reaching  -4 ℃ in February across the whole basin (Figure 12). Maximum positive
anomalies are found in July-August, reaching ~4℃ in the northern part and ~+2-3℃ in the southern part. This implies that the amplitude of the seasonal cycle is larger in the northern than in the southern Red Sea. Minimum negative anomalies with respect to the annual mean in the outer region were found in August (~-2℃) and maximum anomalies (~+3℃) are found in May, with both these anomalies being larger in the open ocean than in the Gulf of Aden.



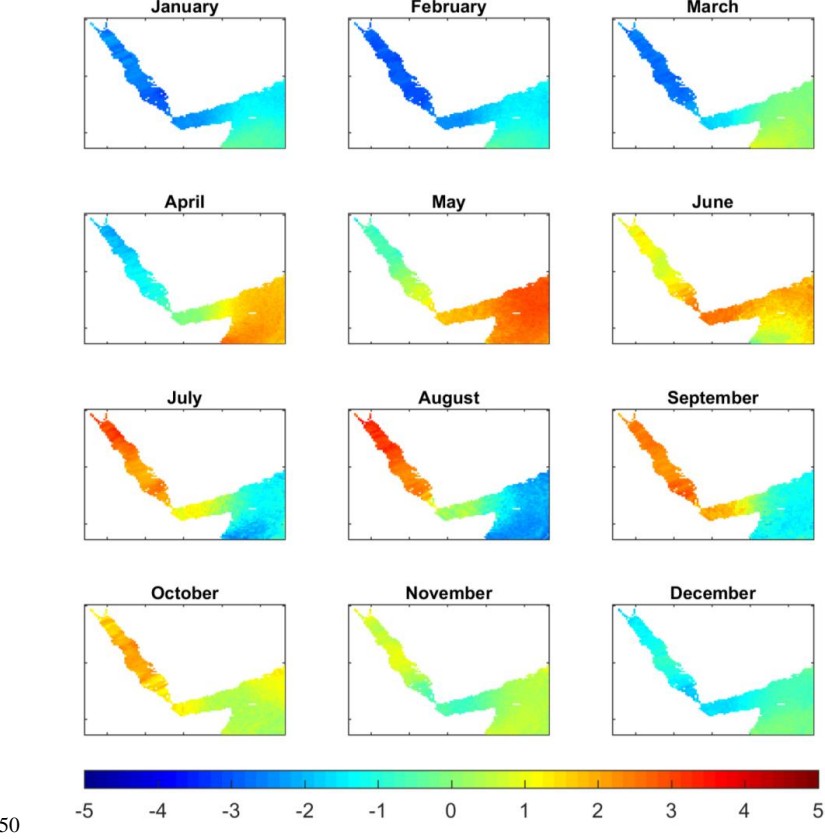

**Figure 12: Monthly climatological anomalies of the surface temperatures with respect to the annual mean (in ºC).**

A seasonal thermal regime is only detected above 80-100 m in the Red Sea, being larger in the shallowest layers. In the Red Sea (Gulf of Aden) the larger negative anomalies are found in February (August) and the larger positive anomalies are found in August (May). The Gulf of Aden presents large seasonal variations between 50 and 200 m, with departures from the annual mean range from -4ºC to +4ºC from August/September to April/May.

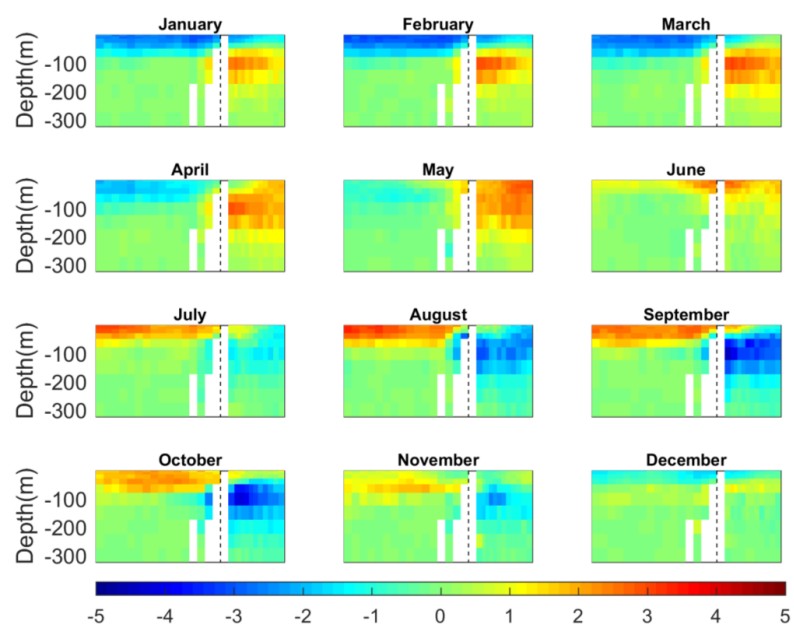

**Figure 13: Monthly climatology of temperature anomalies along the section depicted in Figure 11a, with a zoom in the 0-300 m layer.**

The seasonal evolution of the depth of the thermocline, defined here as the depth showing the maximum vertical temperature gradient, was computed for each grid point and then averaged regionally (Figure 14). In the Red Sea the thermocline was deeper in February and shallower in the summer months, as expected. However, a clear difference is found between the northern and southern regions. In the northern part the thermocline is deeper, reaching 80 m in February, while in the southern part it is rather constant with monthly-averaged values ranging from 35 m to 50 m. In the outer region, the thermocline is deeper with maximum values of 100 m in March-May and minimum values of 70 m in September-October.

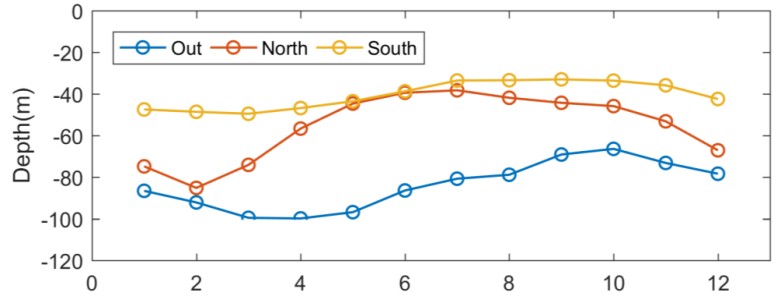

**Figure 14: Seasonal evolution of the regionally averaged depth of the thermocline.**





### 3.3 Interannual variability

In the Red Sea, the std of interannual variations of the basin averaged temperature at the sea surface and
the upper layer (0-100 m) are 0.33 ℃ and 0.34 ℃ respectively (Figure 15), an order of magnitude lower
than the seasonal changes. At the intermediate and bottom layers interannual changes are smaller (0.08 ℃
and 0.04℃, respectively), but in those layers the seasonal variations are negligible, so the relative
importance of low frequency changes is larger. In the outer region, interannual changes are larger in all
layers (Figure16) with a yearly std of 0.38 ℃ and 0.45 ℃ in the sea surface and the upper layer (0-100m),
respectively. In the intermediate layer the interannual std is 0.21 ℃, more than twice larger than in the
Red Sea, probably associated to lateral advection of water masses from the Indian Ocean. In the bottom
layer, the yearly std is lower, 0.05 ℃, but still larger than in the Red Sea.

To characterize if the interannual variability of the temperature field is the same along the year, we have
computed the standard deviation of the time series per months (i.e. 60 values per month; Figure 14). In
the Red Sea, the interannual variations are relatively small, with a std ranging from 0.20℃ in January to
0.70℃ in November, being quite homogeneous along the basin. In the Gulf of Aden, the interannual
variations are larger, particularly from May to November, when std exceeds 1℃. The rest of the year the
std of the interannual variations are smaller (< 0.5℃).





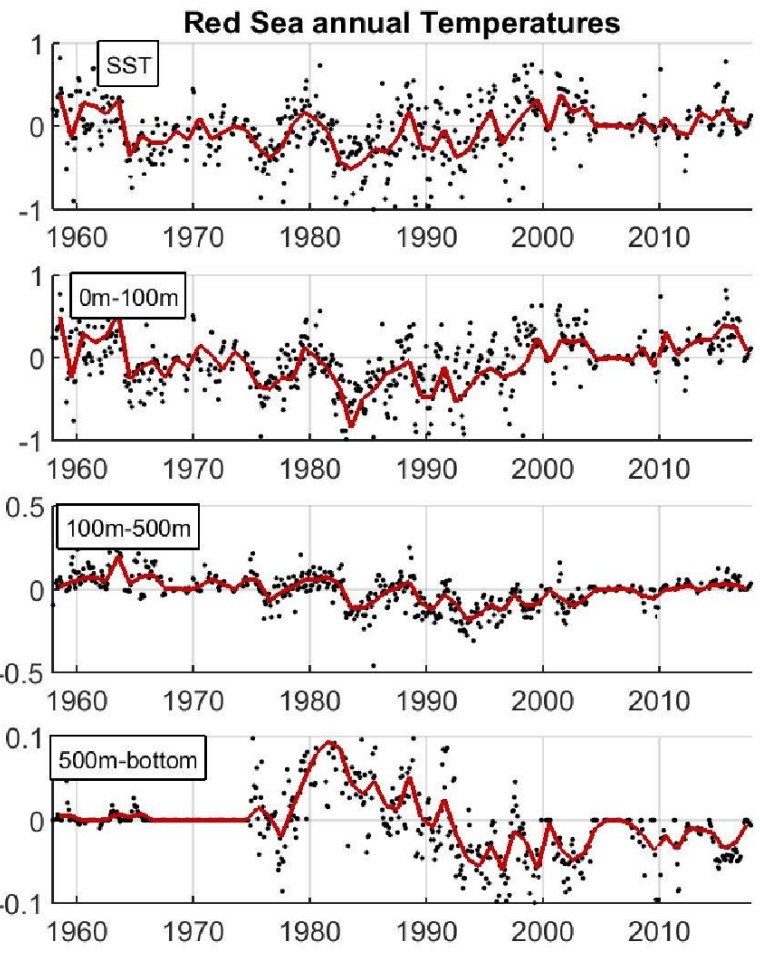

**Figure 15: Time series of yearly averaged temperature (in ºC) in different layers (a) SST, (b) 0-100 m, (c) 100-500 m and (d) 500 m - bottom, in the Red Sea. Black dots indicate the monthly values with formal error below 0.2ºC. Note the different vertical axis in each subplot.**


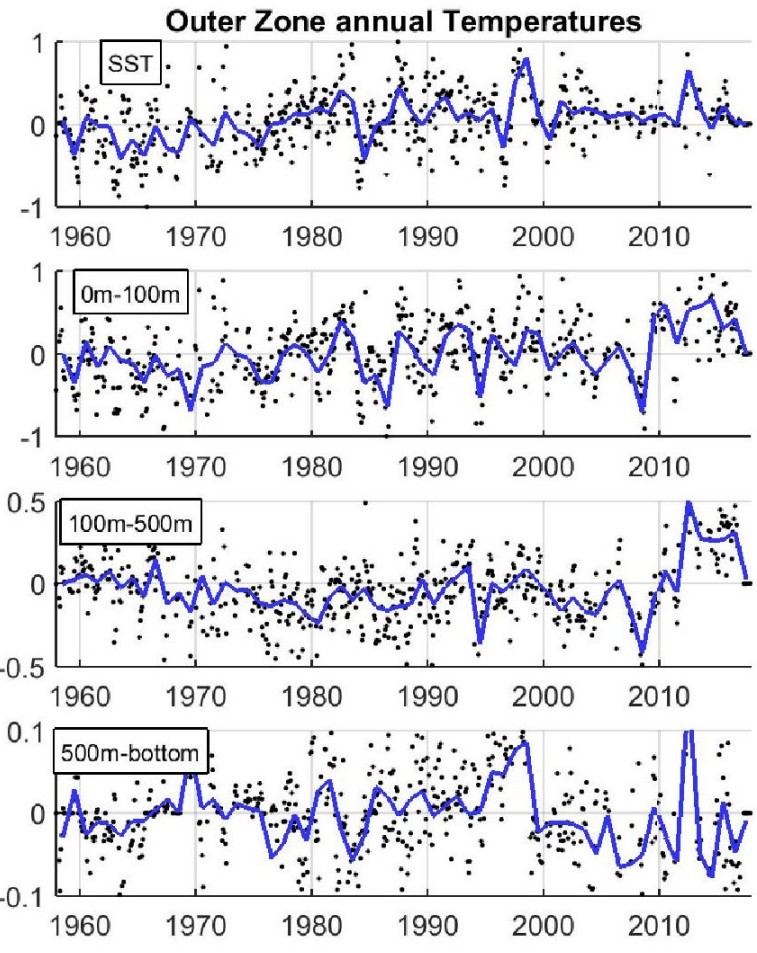

**Figure16: Time series of yearly averaged temperature (in ºC) in different layers (a) SST, (b) 0-100 m, (c) 100-500 m and (d) 500 m - bottom, in the outer region. Black dots indicate the monthly values with formal error below 0.2ºC. Note the different vertical axis in each subplot.**





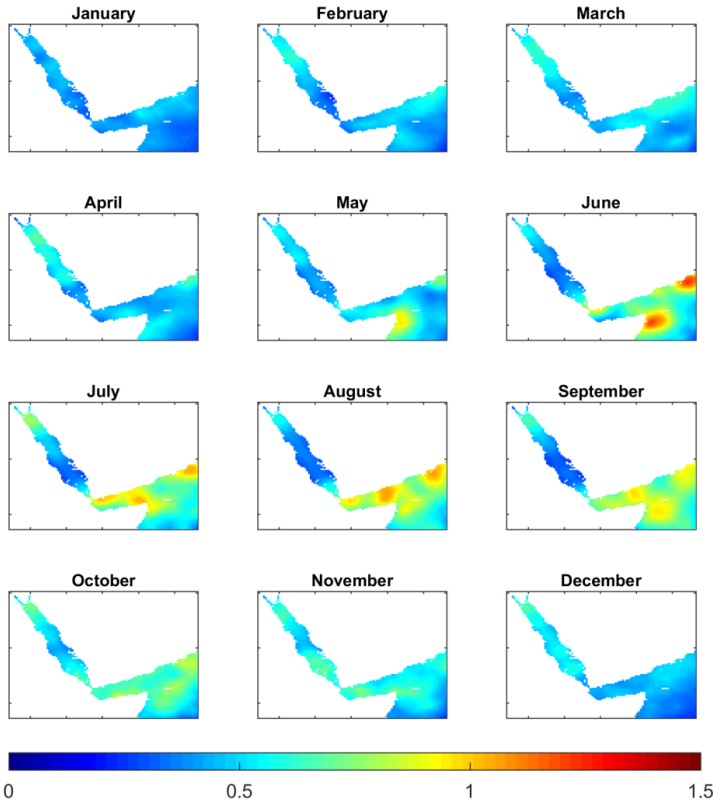

**Figure 17: Std of the interannual variations of surface temperature por months (in °C).**

In the water column, the largest interannual variations are found in the Gulf of Aden, at the same location where the monthly anomalies were the largest, between 50 and 150 m (Figure 18). The std there even exceeds the values in the surface layer, ranging from 1°C in February to up to 2°C in September. Inside the Red Sea, the interannual variability decreases with depth, with a std < 0.1°C below 200 m (Figure 18).





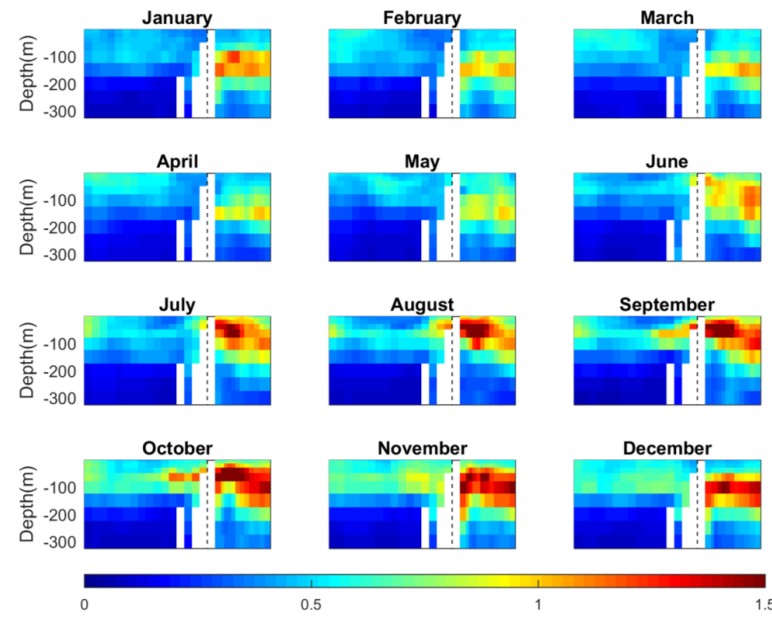


**Figure 18 Vertical section along the Red Sea and Gulf of Aden of the std of interannual variations per months (in ºC).**

### 3.4 Multidecadal changes

The assessment of the long-term changes of the temperature field is of paramount relevance as they can shape the characteristics of the local ecosystems and may help characterize the impacts of global warming in the region. Careful examination of the interannual time series suggests that multidecadal changes are over imposed to the interannual variability. To highlight this, we extract the multidecadal variability applying a moving average with a 10-year window to the monthly time series (Figure 19). In the Red Sea, the low frequency component of the temperature time series in the upper layer show a monotonous

decrease from the 1960's reaching a minimum in mid 1980's increasing monotonically since then. In the late 1960's, the temperatures were similar to those in the present decade, both being ~0.4ºC above the minimum. A similar pattern applies to the intermediate layer, but the minimum was reached a decade later, in the mid 1990's. In this case, the difference between the maximum and the minimum was 0.2ºC, with present temperatures ~0.05ºC below those in the 1960's. In the bottom layer the maximum was

found in the early 1980's, while a minimum was found at the end of the 1990's. The shift in the multidecadal minima may be reflect heat transfer between layers, but available information is insufficient to assess this possibility.

In the outer region, the low frequency component of the temperature in the upper layer shows an almost regular warming since the 1960's, with a relative minimum in the mid 2000's. In the intermediate layer a

more complex behaviour is observed, with two relative minima (in early 1980's and mid 2000's), a relative maximum in the mid 1990's and a clear warming since mid 2000's. The evolution in the deeper layer is similar to the intermediate layer except that no clear warming is observed since the 2000's.

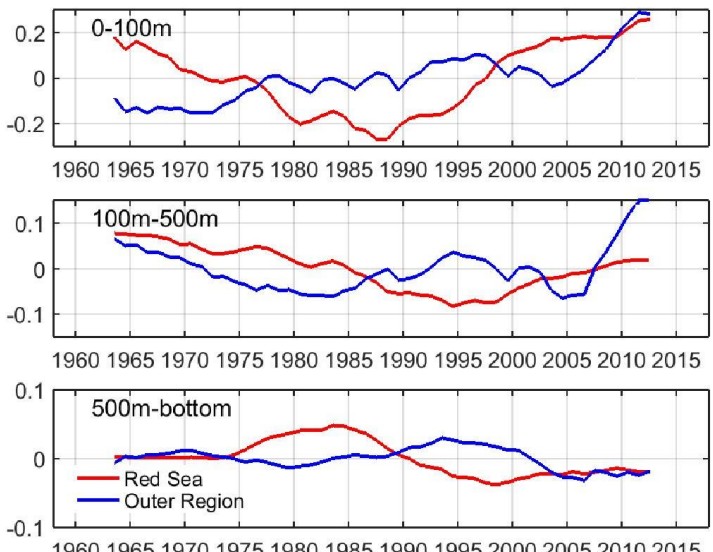

**Figure 19: Low-pass filtered temperature time series (in ºC) at different layers in the Red Sea and the Outer Region. A 10-year moving average has been applied to the monthly time series. Note the different vertical axis in each subplot.**

Finally, we computed the long-term trends in the Red Sea and the outer region at different depths. To do so, we considered that in some months there were few observations, and therefore analysed temperature anomalies were close to 0. So, in order to avoid biases in the trend estimates, months in which the formal error is greater than 0.15ºC are not considered in the computation.

Trends computed for the whole TEMPERSEA time series (1958-2017; Figure 20a), show only positive trends above 40 m depth, with maximum trends of 0.045 + 0.016 ºC per dec at 15 m, and the largest negative trends at 125 m (-0.072 + 0.011 ºC per dec). In the outer region trends are positive in the whole water column, except between 100 m and 250 m. Maximum trends were found at 15 m (0.12 + 0.01 ºC per dec) and the largest negative trends at 175 m (-0.035 + 0.013ºC per dec). For completeness, we also computed the linear trends for the satellite period (1985-2017; Figure 20b). As the period cover by satellite observations includes the recent period of monotonous warming, trends are positive above 250 m, with maximum values found at 50 m depth (0.27 + 0.04 ºC per dec). The largest negative trends are observed at 400 m (-0.04 + 0.01 ºC per dec). In the outer regions, trends are positive above 800 m, with maximum values at 15 m (0.16 + 0.03ºC per dec).





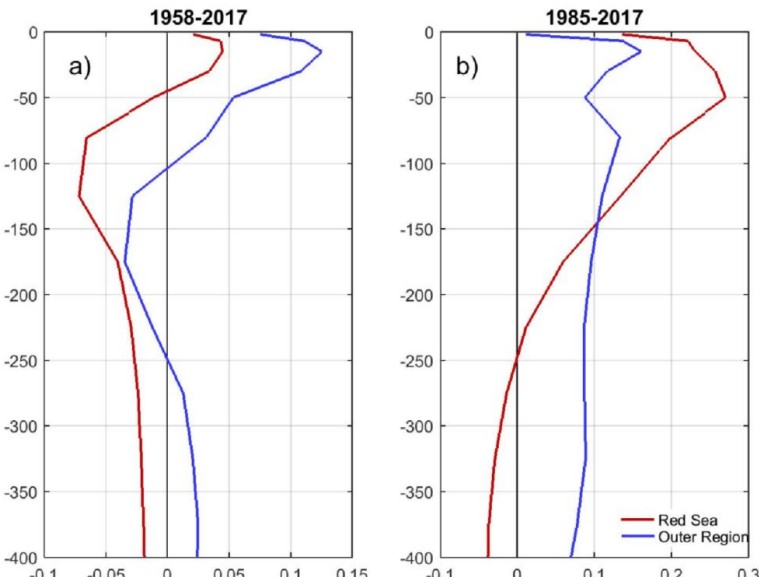

**Figure 20: Vertical distribution of temperature trends (in ºC/decade) for the Red Sea (in blue) and the Outer Region (in red). The linear trends have been computed for the period (a) 1958-2017 and (b) 1985-2017. Note the different horizontal axis in each subplot.**

## 4 Discussion

The TEMPERSEA product provides a homogeneous gridded record of temperature in the whole water column based on quality-controlled in situ observations over the last 60 years. Therefore, it is a valuable complement to the more accurate, but limited on time and depth, satellite-based products. As usual in gridded products, the accuracy of TEMPERSEA is directly linked to the density of in situ profiles, which is rather heterogeneous in space and time. Therefore, us of the TEMPERSEA product should take into account the uncertainty estimates. Our comparison of the formal error estimates and direct estimates based on synthetic experiments suggest that the uncertainty estimates, both at grid point level and for the basin averages, are accurate and are a good indicator of the reliability of the product at a given time/location.

This is especially relevant when long term trends are to be computed. The mapping procedure is a combination of the information provided by the background and by the observations. In cases when/where no observations are available the analysis tends to the background information, which is a monthly climatology that does not change from year to year. This fact artificially damps the estimates of long-term trends (e.g. Llasses, Jordà, & Gomis, 2015), so a careful treatment is needed. Our approach has been to compute trends using only those months that have enough observations (i.e. identified as those with formal error below a certain threshold). Alternatively, (Good et al., 2013), use the analysis of the precedent month as the background field. This allows the propagation of long-term changes and may produce a better estimate of the long-term trends. However, it also may induce spurious trends if sustained periods without observations exist (i.e. several years), so this approach should be careful explored in future analyses of TEMPERSEA.

Another interesting feature of the uncertainty estimate is that it allows identification of sampling strategies that have led in the past to high accuracies, and thus that could be used to guide future monitoring efforts of Red Sea temperatures. The formal error decreases with the number of observations, but the spatial distribution of the observations also plays an important role. In TEMPERSEA more than



70 months in which the error is as low as 0.1ºC with less than 10 observations have been identified (Figure 21). Conversely, in some months with intensive campaigns more than 500 profiles have been collected, but the formal error did not decrease further. The reason for this is that observations separated less than the typical correlation length scale (i.e. the spatial scale of the process dominating the temperature variability) provide redundant information. On the other hand, we have identified months in which, surprisingly, no observations were gathered in the Red Sea. As mentioned before this represents a serious limitation to accurately quantify long term changes. Therefore, if the goal is to characterize the climatic evolution of the Red Sea temperature an optimized sampling should be designed to minimize the number of required profiles, with approximately less than 10 profiles needed per month. However, this should be repeated monthly, or at least seasonally, to ensure the continuity of the record and to reduce the noise in the long-term change estimates.

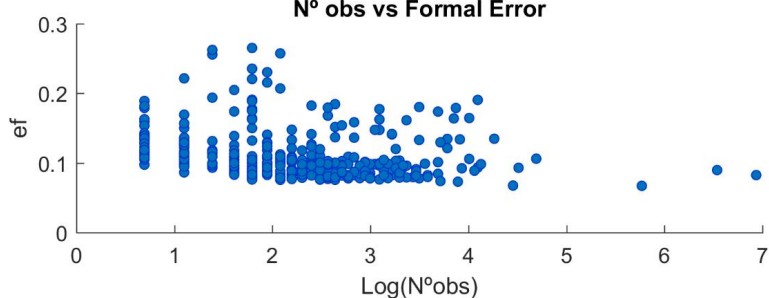

**Figure 21: Scatter plot of the formal error vs log of the number of observations per month used to compute the maps.**

TEMPERSEA has allowed to characterize the 3D variability of the temperature field in the Red Sea and the adjacent Arabian Sea, which show a different behavior. In the Red Sea most variability is induced by surface processes with little variability at intermediate or deep layers. Conversely, in the Gulf of Aden and the Arabian Sea the influence of lateral advection seems to play an important role in inducing a shift in the seasonal cycle and large interannual variations in subsurface layers. In order to get a deeper insight in the role of the atmosphere in the sea temperature variations, we analysed air temperatures at 1000 mbars (representative of air in contact with sea surface) and 850 mbars (representative of air masses not directly affected by air-sea interactions). In particular we used the output from the JRA55 atmospheric reanalysis for the period 1958-2014 (HARADA et al., 2015), and extend it with the output from the NCEP reanalysis (Kanamitsu et al., 2002) for the period 2014-2017. Before merging both datasets we ensured homogeneity in terms of mean and variance during the common period. The air temperatures have been averaged over the Red Sea and the outer region and compared with the sea temperatures at different layers. In order to isolate the interannual variations we have removed the multidecadal variations using a 10-year moving average high-pass filter.

In the Red Sea the results show a very good correlation between air temperature at 1000 mbar and temperatures at the sea surface and in the 0-100 m layer (correlations of 0.78 and 0.81, respectively; see Figure 22 and Table 2). When air temperature at 850 mbar is used, the correlations decrease but still are high (0.68 and 0.69, respectively). This means that most interannual variations in the upper layer of the Red Sea can be explained by large scale changes in air temperature. A non-negligible part (~15% of the variance) can be attributed to air-sea interactions. The effects of atmosphere variability are also detected in the intermediate layer, where the correlation is 0.45. no statistically significant correlations were found in the deeper layers. Concerning the Gulf of Aden, the correlation between air and sea temperatures is lower and restricted to the upper layer (see Table 2). This reinforces the hypothesis that lateral advection plays an important role in driving the interannual variations of temperature in the Gulf of Aden and the Arabian Sea.





| | Red Sea | | Outer Zone | |
|---|---|---|---|---|
| | T air 1000 mbar | T air 850 mbar | T air 1000 mbar | T air 850 mbar |
| Sea Surface | 0.78 | 0.68 | 0.57 | 0.47 |
| Sea 0-100m | 0.81 | 0.69 | 0.43 | N/S |
| Sea 100-500m | 0.45 | 0.37 | N/S | N/S |
| Sea 500 - 1000m | N/S | N/S | N/S | N/S |

**Table 2: Correlation between air and sea temperatures in the Red Sea and the Outer region. Only years with averaged formal error below 0.15ºC are considered. All values are significant at the 95% level (N/S indicated otherwise).**

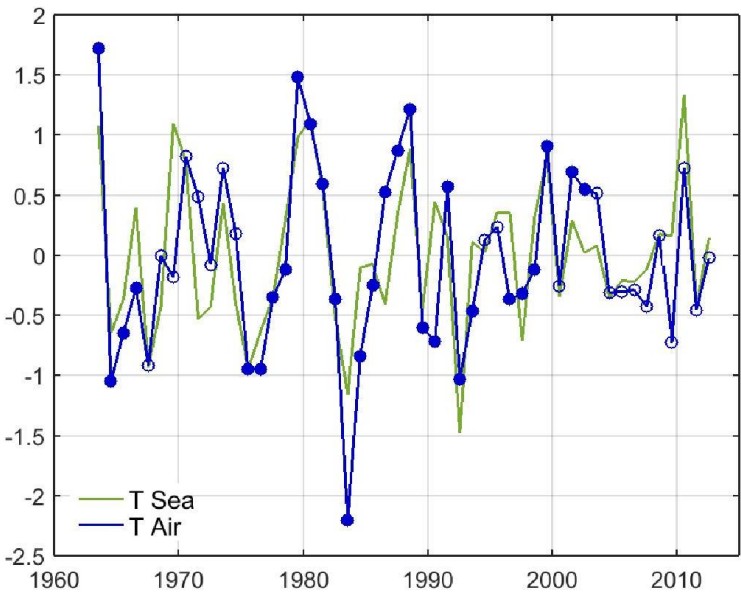

**Figure 22: Normalized air temperature at 1000 mbars (green) and 0-100m sea temperature (blue) in the Red Sea. A high-pass filter has been applied to remove multidecadal variations. Solid dots indicate an averaged formal error below 0.15ºC.**

Finally, multidecadal changes have been assessed with the TEMPERSEA product showing a non-negligible contribution to temperature variability. This is an important result for the interpretation of long
term trends. Linear trends are often computed as an indicator of potential influence of global warming. However, the trends can be masked by low frequency variations when their period is comparable to the length of the record (Jordà, 2014). This is clear for the temperature records in the Red Sea derived from the TEMPERSEA product. For instance, our results suggest that sea temperature in the upper layer, in the 1960's was similar to the present values, so a very small positive trend is obtained when the period 1958-
2017 is used, consistent with recent evidence of long-term thermal oscillations in the Red Sea (Krokos et al., 2019). For the intermediate layer the sign of the trend is even reversed, as the temperatures in the 1960's were higher than those recorded in the recent years. Conversely, if only the last 30 years are considered, which is also the period covered by the satellite record, trends are strong and positive, and


therefore easily misinterpreted as linked to global warming. Hence, the conclusions, based on satellite records, that the Red Sea is warming at rates faster than the global ocean (Chaidez et al., 2017; Raitsos et al., 2011), based on the satellite record, need be reconsidered, as warming rates retrieved for 1958-2017 with the TEMPERSEA product are 10 fold lower than those retrieved from satellite records covering the past 30 years.

## 5 Conclusions

An observational based high resolution and homogeneous 3D temperature product has been developed for the Red Sea for the period 1958-2017 (TEMPERSEA product). For that, two databases of in-situ observations (CORA and KAUST) were merged and quality-controlled, resulting in a dataset of 41713 profiles (11191 in the Red Sea and 30522 in the Gulf of Aden). A mapping procedure based on optimal interpolation has been applied to those profiles to compute two gridded products: a 12-month climatology 545 and a 60-year monthly product. In order to calibrate the algorithm, synthetic data from a realistic numerical model have been used. Furthermore, the formal error from optimal interpolation have been computed and has proven to be a good approximation to actual uncertainty. The TEMPERSEA product is available from the open data repository Pangea (Agulles et al. 2019[1]).

The product has been compared to satellite observations for the period 1981-2017 showing reasonable 550 agreement in terms of spatial and temporal variability at monthly, seasonal and interannual scales. Also, very good agreement has been found between air temperatures from the atmospheric reanalyses and reconstructed sea temperatures for the whole period 1958-2017, enhancing the confidence on the quality of the product.

The TEMPERSEA product allowed us to characterize the climatology of the temperature in the region. In 555 the Red Sea, the maximum temperatures are found south of 20°N, while the minimum is found in the northern part. Regarding to the seasonal cycle, it peaks in August and is minimum in February. The seasonal cycle is larger in the northern part while in the southern part is smaller in terms of the thermal range in surface waters. In the Gulf of Aden, the phase and shape of the seasonal cycle is different with maximum values in May and minimum values in August. Related to the vertical structure of the 560 temperature field, our results show a large difference between the Red Sea and the Gulf of Aden, specially below the depth of the Bab-El-Mandeb Strait. The Strait isolates the Red Sea allowing it to have temperatures above 20ºC in the whole water column, while the Gulf of Aden, influenced by the open ocean variability show a vertical structure typical of the Indian Ocean, with temperatures reaching 5ºC at 1000 m depth. Furthermore, the length of the product has allowed to characterize multidecadal variability 565 at different layers. Our results show that multidecadal variations have been important in the past and can bias high the trends computed from 30-40 years of data.

TEMPERSEA provides a reference product to describe the temporal evolution of the 3D temperature field in the Red Sea and to calibrate/validate numerical models. This will allow to improve forecasting models and formulate more reliable predictions and climate projections. It has also been shown that the 570 quality of the product is critically linked to the existence of in situ observations. Periods with few observations degrade the quality of the product, so it is important to keep a regular monitoring of the region in order to identify new changes and to remove uncertainties in the climate studies. TEMPERSEA provides a basis to design an optimal sampling program to track the thermal dynamics of the Red Sea. Our results suggest that an effective monitoring can be achieved with few, strategically located, 575 observations.

---

[1] The data set will be published in Pangea.de upon acceptance of the manuscript





**Acknowledgements**

This research was funded by King Abdullah University of Science and Technology (KAUST) through
funds provided to S.A. and C.M.D (BAS/1/1072-01-01).

M.A. has been partly funded by the European Union's Horizon 2020 research and innovation programme
under grant agreement No 776661 (SOCLIMPACT project).

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
