# Peer review of "Temporal evolution of temperatures in the Red Sea and the Gulf of Aden based on in-situ observations (1958-2017)"

_Ocean Science, 2019_

## Referee Comment (RC1) · Anonymous Referee #1 · 27 Aug 2019

This work analyzes a large data set of temperature profiles obtained in the Red Sea from 1958 to 2017. The data sources are several data collections. The analyses are differentiated for three different areas: The northern and the southern Red Sea, and an outer area to the east of the Bad-el-Mandeb Strait. The analyses presented are quite exhaustive and include a description of the quality control process, the data interpolation method, and an inter-comparisons with model and SST satellite data. First the seasonal cycle of temperature for the different regions and depth ranges are analyzed and then the inter-annual and multidecadal variability is addressed.

In my opinion this work is very exhaustive and interesting. The main objectives of establishing the seasonal cycle of temperature as a function of the geographical location (Longitude/attitude) and depth, and studying the time variability at inter-annual

and multidecadal scales are achieved. The manuscript is well organized and, in general terms, well and clearly written. For all these reasons I believe it is suitable for publications with minor revisions.

My main concern is the lack of an analysis of the salinity data. I assume that many of the available profiles analyzed come from CTD profiles or Argo profilers and therefore salinity data are also available. The analysis of temperature is very interesting by itself, but it would be much more complete if the companion salinity information was included. Note that the Red Sea is one the places of the world ocean with a highest evaporation and therefore the salinity variability and possible alterations could be of paramount importance. Furthermore, the dynamics of the circulation of the Red Sea would be driven by the density field (despite the wind-driven circulation). If the temperature changes are compensated by salinity changes then the density field is not altered. I think it would be important to know if this is happening or not. I am not an expert in the Red Sea circulation, but as long as I know, there is a thermohaline circulation and a water exchange with the Indian Ocean in order to compensate for the strong evaporation. Once again this depends on the density field and the joined action of temperature and salinity. Nevertheless, I understand that the role of the reviewer is to review the present work, not to suggest a different work. For this reason I consider this as a minor concern. The analysis of the temperature data merits publication by itself and I simply suggest that including a salinity analysis would improve very much the work.

Other minor points. Introduction. Figure 1. For those people not familiarized with this region, a figure from a wider geographical area should be included in order to locate the Red Sea. Then, the present figure 1 could be a zoom from the larger area. At the beginning of the introduction (first paragraph), I miss a description of the Bad-al-Mandeb strait, mainly its maximum depth which I guess conditions the exchange between the Red Sea and the Indian Ocean. Otherwise, the introduction is clear and informative.

Line 107: "the data has been quality controlled...". It is true that the quality control is explained later in section 2.4, but the first time I read it I wondered how had been done the quality control?. Please, include a parenthesis (see section 2.4) for impatient readers like me.

Lines 116 and 117. This is the first time that OSTIA and ICOADS appear. Have this acronyms an explanation? Please, include it.

Line 120: "Both OSTIA products are merged after a cross validation is performed". What kind of cross validation? How was it carried out? Please, explain it just a little.

Line 143: "...to remove spikes, out layers and density inversions". It is clear what a density inversion is, but the criteria to determine if a data point is an out layer is more subjective. Which criterion was been followed: two standard deviations from the mean value?, three?, those values beyond a certain percentile? Is the procedure the one explained in lines 150-155, or this is a different quality control? Why you use the 1% and 99% percentile criterion in some cases and the three standard deviations in other cases?

Lines 185-190. I do not like very much these sentences. In Optimal Interpolation, the observations are considered as composed by a background field, a signal and an error, which is not necessarily a measurement error, but simply the part of the observation corresponding to a length scale on which we are not interested. The interpolated values are estimated using the statistics of the signal (variance and decaying scale) and the signal/error ratio. So I believe that "the weights are determined from the statistics of the observational errors" is not a good description.

Expression (1) could be improved. When writing in the left had side of the equation $V(r)$ it seems to me that it is the value of variable $V$ at the coordinate vector $r$ (you say at a "given position $r$"). Then you say that $BK$ is a M-vector. In that case $V$ is also a vector, or $r$ is a vector of positions.
In expression (5) $T_{ij}/T$, I guess the exponent should be negative in the same way the exponent for the spatial correlation is negative. Otherwise the correlation increases with time.

Figure 12. I would represent directly the values of the temperature for the climatology. In that way you would know the temperature for each month of the year for the climatological cycle. In the present way, you have to look at the mean temperature and then add the anomaly. It is not very helpful. In line 345 and followings it is stated that the minimum anomaly for the seasonal cycle, and then the minimum temperatures along the year (it would be better to see temperatures directly) are found in August in the outer part. Taking into account that this area in to the north of 10° N, therefore in the northern hemisphere, it seems strange to the reader not familiarized with this region of the world that the minimum temperatures are reached in August, when one expect the maximum ones in the northern hemisphere. I think that this result needs some more explanations for the non-expert readers like me.

You compare sea temperature with air temperature at 1000 mbars, considered as the air in contact with the sea, and at 850 mbars. I think that using 850 mbar temperature makes no sense. The heat exchange between the sea and the atmosphere depends on the temperature of the air above it. If the air at 850 mbar is very warm, but the air at the sea surface is cold, the cold air would enhance latent heat and sensible heat fluxes, no matter which is the temperature at 850 mbar. A different question is that 1000 and 850 mbar temperatures are very likely to be correlated, and therefore sea temperature and 850 mbar temperature are also correlated. My point is that we should not use time series to calculate correlations just because such time series are available. There must be some scientific reason. If you already have 1000 dbar temperature, please, do not use 850 dbar. It gives the false impression that there is some sort of phenomenon that can influence the sea temperature from the upper part of the atmosphere.

In line 373 you use the abbreviation std. I suppose it means standard deviation. Please, define it previously.
Some writing errors. Line 442: "the period cover by..." should be covered. Line 546. "the formal error from optimal interpolation have..." should be "has".

---

## Short Comment (SC1) · 5 Sep 2019

Interactive comment on the work entitled "Temporal evolution of Red Sea temperatures based on in situ observations (1958–2017)" is listed below and attached as a file along with this. (by C P Abdulla).

Appreciating the authors for the work entitled "Temporal evolution of Red Sea temperatures based on in situ observations (1958–2017)" by Miguel Agulles et al., 2019 which has analyzed the in situ profiles in the region and developed a gridded product based optimal interpolation technique. The article further discussed the seasonal, interannual and decadal signal in the temperature of the Red Sea and outer region (mainly Gulf of Aden). My major concern is on the analysis and some of them are listed below.

Comment 1: Please add in the text about the criteria used for removing the spikes, out layer and density inversion. Comment 2: In Figure 2, why is the left panel the out data are plotted, it would be better to keep only the Red Sea data to cope with the caption of the Figure. Comment 3: Figure 5 shows the distribution of all the available observations for January in the region and the profiles distribution after applying the K-means algorithm. Are these profiles shown in (Figure 5b) the only profiles used in Optimal Interpolation? Comment 4: When I check the data availability in the Red Sea region from World Ocean Database, the data points are mostly aligned along the center with significantly lower number profiles towards both eastern and western coast. To what extent the second source of data cover this in space and time? Comment 5: The 3D gridded temperature product spanning for the period 1958-2017 is will be very helpful in understanding the Red Sea. From my understanding of the manuscript, I found that the amount of profiles in the Red Sea used for the analysis is very low, except for 2 or 3 years (∼1959, ∼2000 and ∼2016). If this is true, is the derived product will be reliable to discuss interannual and decadal signal? Comment 6: A table explaining the number of profiles used in the OI per each decade separately in the Red Sea will be helpful to show the data distribution in the Red Sea (which is the prime focus of the study) used in the analysis in addition to a map showing the data spread can be added as supplementary file. Comment 7: Most of the data represent the outer region and few only represent the Red Sea, so the title of the manuscript and the name of the product should consider that.

Please also note the supplement to this comment:
https://www.ocean-sci-discuss.net/os-2019-66/os-2019-66-SC1-supplement.pdf

---

## Referee Comment (RC2) · Anonymous Referee #2 · 19 Sep 2019

This study investigates the temperature distribution in the Red Sea from observations collected from 1958 to 2017. The authors combine the data from multiple sources and apply a stringent quality control resulting in a high quality data set which is interpolated to produce a gridded climatology. This allows for an understanding of the Red Sea variability.

As the observational data was collected from CTDs the article could have been greatly improved if the authors had included the analysis of salinity and done the calculations along density isopycnals rather than on depth surfaces. Furthermore the temperature used needs to be either Conservative Temperature or potential temperature not in situ temperature.

I was surprised by the high percentage of the observations data was located incorrectly,

are the authors sure there is not a salinity compensation to this low temperature water that produces an appropriate density for this region.

Overall I found the paper to be well written and is interesting and I believe it should be published. It is great that the authors made TEMPERSEA freely available.

---

## Editor Comment (EC1) · Trevor McDougall (Editor) · 8 Oct 2019

Dear Authors,

You are now in receipt of some very helpful comments from the reviewers.

Please change your manuscript in accordance with these comments, get back to me with a point-by-point summary of how you have handled each comment as you prepare your manuscript for Ocean Science.

---

## Referee Comment (RC3) · Anonymous Referee #3 · 9 Oct 2019

This paper takes generally available in situ temperature profile data for the Red Sea and Gulf of Aden, combines it with newly available data to create long-term climatological mean fields of surface and subsurface temperature as a baseline for time series of month/year temperature fields (surface and subsurface) for all months for years 1958-2017. Error estimates are calculated from subsampled GLORYS reanalysis data. Some discussion of season, interannual, and decadal variability is included, with decadal trends of opposite sign at the surface and at 125 m depth.

This work is definitely of interest, both for the climatological mean fields of temperature in the Red Sea and Gulf of Aden, and for the analysis of seasonal to decadal changes in the temperature field, with their influence on the climate and biota of the region. The authors write very clearly regarding the method used, with a particularly

nice explanation of optimal interpolation and of the calculation of error statistics. A more thorough examination of the data would improve the paper, as would validation of the subsurface long-term mean fields against existing products, and more discussion of results, particularly trends of opposite sign at different levels in the water column. Details below.

First, the addition of the KAUST data set is a welcome augmentation of existing data for the Red Sea, especially with the possibility of continued monitoring by this source. I do not know the data policy for this journal, but the data used within the paper should be publicly available for reproducibility. The authors should note in the paper where the data can be obtained.

Figure 2 shows a rather startling distribution of temperature values in the Red Sea, especially with what appears to be a very large number of profiles with temperatures well outside the range of Red Sea temperatures at deeper depths. It would be a great service if the authors could detail the data a little more especially those which they state must have erroneous positions. This would help users (and maintainers) of CORA and similar data sets to examine and either flag or correct the erroneous data. Did the authors use CORA quality flags? Did these erroneous data have CORA quality flags?

Figure 7 shows a patter of RMSE Glorys vs. climatology (and optimal algorithm) that appears suspicious - with what looks like the exact same pattern in the 1960s, 1980s, and 2000s centered at 1000 m with the intermittent decades showing near zero error. Can the authors explain this? Is it some kind of decadal cycle embedded in Glorys, rendering it maybe less than useful for error analysis? It also might be nice to enlarge the upper few hundred meters where the largest errors are found, but hard to see in the full vertical graphic.

The long-term climatological mean field is discussed at length, but only validated with a comparison with AVHRR at the surface. It should be compared at subsurface depths to the World Ocean Atlas 2018 (WOA18) field, which are on the same grid size (0.25 x

0.25) and over nearly the same time period (1955-2017) - or another widely used long-term global climatological mean field. This comparison could yield some interesting results as to the efficacy of concentrating on a specific region, instead of using a region of a global climatology, with attendant extra attention, quality control, and in this case new data sources.

Grid size - sampling strategy: is a 0.25 x 0.25 degree grid really necessary to capture temperature change in the Red Sea? According to the authors discussion, less than 10 temperature profiles per month are necessary to adequately quantify temperature change in the Red Sea. If that is truly the case, would not a 1.0 x 1.0 grid along the axis of the Red Sea be sufficient to capture temperature change? In figure 4, it is very hard to see the grid structure used - is there another way to represent it? Maybe just in black and white rather than color? But assuming there are multiple grids laterally across the Red Sea at each latitude, it appears that the K-mean algorithm aggregates data into one or sometimes two grid areas across the Sea longitudinally (Figure 5). These appears to lose any advantage of a 0.25 x 0.25 grid resolution. It may be due to the graphic, but the authors should spend some more time discussing the importance of the 0.25 x 0.25 grid resolution to this work.

Time frequency: similarly, what is the advantage of the month/year time frequency (12 monthly temperature fields in the Red Sea per year 1958-2017)? As the authors note (with the term "surprisingly" though I dont think it should be surprising to the authors who are familiar with historic measurement strategies in the Red Sea) there are many months without any data at all in the Red Sea, and other months with very few measurements. Seasonal temperature cycle in the Red Sea is examined from a climatological (long-term) perspective. I dont see any particular explanatoin of the advantage to month/year fields over simple yearly fields in quantifying and discussing interannual and decadal variability, especially for data sparse years. The authors should do a little more explanation of why monthly fields are produced. At the least a matrix of coverage (or lack thereof) for each month/year should be presented graphically. This would give

a better understanding of data sparsity influence on error, as a companion to figure 21.

In discussing results, the authors note that most interannual variations in the upper layer of the Red Sea can be explained by large scale changes in the air temperature. This is not completely convincing. There is a good correlation, but isnt it equally as likely that it is the air temperatures influenced by the upper ocean temperature rather than the other way round? Short wave radiation as well as trapped long wave radiation is absorbed by the ocean surface and radiated back at a slower rate to the lower atmosphere. A little more discussion would be needed to convince that it is large scale air temperature which is the major factor in the upper ocean.

One of the remarkable features the authors find is that upper ocean temperatures are increasing (decadally) but lower depths are decreasing. How can this be if the main factor in the temperature change is air temperature, and there is little exchange with any water source outside the Red Sea? It may be that the answer has to do with the interannual change in the depth of the thermocline. Figure 14 shows thermocline depth seasonal change. Thermocline depth in the south is fairly constant over the year, but changes in the north. If the thermocline were to shallow in February say, cooler water would be higher in the water column and heating would be concentrated closer to the surface, creating the opposite sign trend pattern with depth shown by the authors. This is speculation, but it would be worth a bit more investigation by the authors to validate and maybe explain the change in sign for decadal trend.

Small things

- line 98: what does "delayed mode" mean here? - lines 116, 117, if OSTIA and ICOADS are acronyms, they should be defined. - line 132, "sea-ice concentration" maybe could be removed. GLORYS may assimilate but it is irrelevant in the Red Sea. - it would be nice, in figure 3 to give some indication of the data which came from KAUST as opposed to CORA. - line 204, add space between "as" and gamma. - line 206, "pof" should be "of" - lines 315-317, why wold satellite data from the top mm of the water

column have a larger variability than in situ data from 2-4 m? - line 412, "imposed to" should be "imposed on" - lines 493-494, lateral advection seems to play an important role..." replace "seems to play" with "plays" if there is actual evidence for this. - line 561, "specially" should be "especially" - lines 565-566, "Our results show that multidecadal variations have been important in the past and can bias high the trends from 30-40 years of data." How can multidecadal trends, presumably a cycle, bias high trends? Are the authors referring to multidecadal trends which are not fully represented in 30-40 years? It appears from figure 19 that this could be so in this specific case, but as a generality a partial cycle could bias trends either high or low. Authors should either remove "high" or refer specifically to the Red Sea trend.

---

## Editor Comment (EC2) · Trevor McDougall (Editor) · 9 Oct 2019

Dear Authors, 9 October 2019

There has been n additional, late review of your paper. This review also makes some very valid points, and you should take this on board during the revision of this paper for Ocean Science.

Many thanks,

Trevor McDougall

---

## Author Comment (AC1) · 5 Nov 2019

This work analyzes a large data set of temperature profiles obtained in the Red Sea from 1958 to 2017. The data sources are several data collections. The analyses are differentiated for three different areas: The northern and the southern Red Sea, and an outer area to the east of the Bad-el-Mandeb Strait. The analyses presented are quite exhaustive and include a description of the quality control process, the data interpolation method, and an inter-comparisons with model and SST satellite data. First the sea-

sonal cycle of temperature for the different regions and depth ranges are analyzed and then the inter-annual and multidecadal variability is addressed. In my opinion this work is very exhaustive and interesting. The main objectives of establishing the seasonal cycle of temperature as a function of the geographical lo- cation (Longitude/attitude) and depth, and studying the time variability at inter-annual C1

and multidecadal scales are achieved. The manuscript is well organized and, in general terms, well and clearly written. For all these reasons I believe it is suitable for publications with minor revisions.

- We deeply thank the referee's comments and the effort he/she made in carefully reviewing our work. In the new version of the manuscript we have implemented all the points raised in the review. Thanks to those advices, the new version of the manuscript has been improved.

My main concern is the lack of an analysis of the salinity data. I assume that many of the available profiles analyzed come from CTD profiles or Argo profilers and therefore salinity data are also available. The analysis of temperature is very interesting by itself, but it would be much more complete if the companion salinity information was included. Note that the Red Sea is one the places of the world ocean with a highest evaporation and therefore the salinity variability and possible alterations could be of paramount importance. Furthermore, the dynamics of the circulation of the Red Sea would be driven by the density field (despite the wind-driven circulation). If the temperature changes are compensated by salinity changes then the density field is not altered. I think it would be important to know if this is happening or not. I am not an expert in the Red Sea circulation, but as long as I know, there is a thermohaline circulation and a water exchange with the Indian Ocean in order to compensate for the strong evaporation. Once again this depends on the density field and the joined action of temperature and salinity. Nevertheless, I understand that the role of the reviewer is to review the present work, not to suggest a different work. For this reason I consider this as a minor concern. The analysis of the temperature data merits publication by

itself and I simply suggest that including a salinity analysis would improve very much the work.

- Thanks for the comment. We also believe that salinity is important, but there have been several reasons for us to not include its analysis in this work. The number of salinity observations in the basin is significantly smaller than the temperature. At the same time, the correlation length scales for salinity are smaller than those of temperature (Llases et al, 2016), so more data would be required to obtain a reliable product. Additionally, including salinity would require specific tests to calibrate the algorithm, and to quantify the uncertainties, which would involve a huge extra effort. For all this, we have prefered to focus on temperature characterization, specially considering that temperature has been recognized as the most influential factor for Red Sea ecosystems. We hope in the near future there will be enough salinity profiles thanks to the new observational systems that will allow us to produce an equivalent product for the salinity.

Other minor points. Introduction. Figure 1. For those people not familiarized with this region, a figure from a wider geographical area should be included in order to locate the Red Sea. Then, the present figure 1 could be a zoom from the larger area.

- This figure has been modified in the paper.

At the beginning of the introduction (first paragraph), I miss a description of the Bad-al-Mandeb strait, mainly its maximum depth which I guess conditions the exchange between the Red Sea and the Indian Ocean. Otherwise, the introduction is clear and informative.

- This information has been added in the first paragraph of the paper (L43).

C2 Line 107: "the data has been quality controlled. . .". It is true that the quality control is explained later in section 2.4, but the first time I read it I wondered how had been done the quality control?. Please, include a parenthesis (see section 2.4) for impatient

readers likeme.

- Thanks for the suggestion. The parenthesis has been included in the revised article.

Lines 116 and 117. This is the first time that OSTIA and ICOADS appear. Have this acronyms an explanation? Please, include it. - This has been updated in the revised article.

Line 120: "Both OSTIA products are merged after a cross validation is performed". What kind of cross validation? How was it carried out? Please, explain it just a little.

- That mergins is done by the OSTIA team. In particular, the cross validation of both OSTIA products is done estimating the bias in each product by calculating match-ups between each product and a reference data-set. The details of the procedure can be found in (Bell et al., 2000). This explanation has been added to the text (L121).

Line 143: ". . .to remove spikes, out layers and density inversions". It is clear what a density inversion is, but the criteria to determine if a data point is an out layer is more subjective. Which criterion was been followed: two standard deviations from the mean value?, three?, those values beyond a certain percentile? Is the procedure the one explained in lines 150-155, or this is a different quality control? Why you use the 1% and 99% percentile criterion in some cases and the three standard deviations in other cases?

- The paragraph that explain this part has been modified to better explain the quality control process. The quality control has been done in three steps: Firstly, spikes and profiles with density inversions have been removed in all the area studied (Red Sea and outer region). Secondly, those profiles in the Red Sea showing temperatures colder than 20°C below 500 m have been removed. This has been done because no temperature below 20°C has been found in the reference KAUST dataset at any depth. Finally, as a third step, for the rest of the profiles (in the Red Sea and outer Region), those lying outside a range defined by three times the standard deviation are

also rejected. The 1% and 99% are used just for visualization of the range of values in the reference dataset, which have helped to identify the 20°C threshold mentioned about. This has also been clarified in the text.

Lines 185-190. I do not like very much these sentences. In Optimal Interpolation, the observations are considered as composed by a background field, a signal and an error, which is not necessarily a measurement error, but simply the part of the observation corresponding to a length scale on which we are not interested. The interpolated values are estimated using the statistics of the signal (variance and decaying scale) and the signal/error ratio. So I believe that "the weights are determined from the statistics of the observational errors" is not a good description.

- In the original formulation of Optimal Interpolation (e.g. Gandin et al 1965) the weights of the background and the observations are defined in terms of the covariances of the background and observational errors. In the application of OI to atmosphere/ocean data those error covariances cannot be measured so they are defined using analytical formulations that involve a decay scale (e.g. Gaussian functions), and the error variance is substituted by the field variance. We agree that the original sentence in the manuscript was rather vague and we have corrected it. Now it reads: "OI is an algorithm that estimates the optimal value of the field as a linear combination of available observations and a background (i.e. first guess) field, with weights determined from the covariances of observational and background errors"

Expression (1) could be improved. When writing in the left had side of the equation V(r) it seems to me that it is the value of variable V at the coordinate vector r (you say at a "given position r"). Then you say that BK is a M-vector. In that case V is also a vector, or r is a vector of positions.

- The reviewer is right. In the left side of the equation 1 we have removed (r). The left side represents the analysed field which is a vector, not just a point in a given position.

In expression (5) Tij/T, I guess the exponent should be negative in the same way the

exponent for the spatial correlation is negative. Otherwise the correlation increases with time, The reviewer is right.

- We have corrected it. Thanks.

Figure 12. I would represent directly the values of the temperature for the climatology. In that way you would know the temperature for each month of the year for the climatological cycle. In the present way, you have to look at the mean temperature and then add the anomaly. It is not very helpful. In line 345 and followings it is stated that the minimum anomaly for the seasonal cycle, and then the minimum temperatures along the year (it would be better to see temperatures directly) are found in August in the outer part. Taking into account that this area in to the north of 10âŮęN, therefore in the northern hemisphere, it seems strange to the reader not familiarized with this region of the world that the minimum temperatures are reached in August, when one expect the maximum ones in the northern hemisphere. I think that this result needs some more explanations for the non-expert readers like me.

- Before initial submission of the paper we had discussed a lot about how to present the seasonal variations. We had prepared both figures (for absolute values and for anomalies) and we had no clear preference as both options have pros and cons. Following the suggestion of the reviewer we have modified Figure 12, so it shows the absolute values. Regarding the minimum values observed in the Gulf of Aden in summer, they are caused by the advection of cold waters from the Indian Ocean. The description of the detailed mechanism introducing that advection is out of the scope of the paper. Nevertheless we have introduce a sentence in the manuscript (L360) that reads: "These results suggest that the relative minimum found in the Gulf of Aden during summer could be induced by the advection of cold waters from the Indian Ocean."

You compare sea temperature with air temperature at 1000 mbars, considered as the air in contact with the sea, and at 850 mbars. I think that using 850 mbar temperature makes no sense. The heat exchange between the sea and the atmosphere depends

on the temperature of the air above it. If the air at 850mbar is very warm, but the air at the sea surface is cold, the cold air would enhance latent heat and sensible heat fluxes, no matter which is the temperature at 850 mbar. A different question is that 1000 and 850 mbar temperatures are very likely to be correlated, and therefore sea temperature and 850 mbar temperature are also correlated. My point is that we should not use time series to calculate correlations just because such time series are available. There must be some scientific reason. If you already have 1000 dbar temperature, please, do not use 850 dbar. It gives the false impression that there is some sort of phenomenon that can influence the sea temperature from the upper part of the atmosphere.

- We appreciate your opinion, and we try to explain here our point. The 1000 mbar temperature is the one in contact with the sea, but it is well acknowledged that the sea temperatures also modify air temperatures at the air-sea interface. Therefore, correlations between SST and 1000 mbar temperatures could be due to oceanic effects on the atmosphere. That is the reason why we decided to use the 850 mbar temperatures, not because there were correlations. With that variable we intend to characterize the temperature of the air masses not affected by the air-sea interactions, as stated in the text (L508). By doing this, it is easier to interpret the correlations found: the changes in the temperature of the air (i.e. advection of air masses) is what drives the temperature in the Red Sea.

In line 373 you use the abbreviation std. I suppose it means standard deviation. Please, define it previously.

- Done. C4

Some writing errors. Line 442: "the period cover by. . ." should be covered. Line 546. "the formal error from optimal interpolation have. . ." should be"has".

- This has been corrected Interactive comment on Ocean Sci. Discuss., https://doi.org/10.5194/os-2019-66, 2019.

Please also note the supplement to this comment:
https://www.ocean-sci-discuss.net/os-2019-66/os-2019-66-AC1-supplement.pdf

————————————————————

---

## Author Comment (AC2) · 5 Nov 2019

Cheriyeri Poyil Abdulla abducps@gmail.com
Miguel Agulles et al., 2019 which has analyzed the in situ profiles in the region and developed a gridded product based optimal interpolation technique. The article further discussed the seasonal, interannual and decadal signal in the temperature of the Red Sea and outer region (mainly Gulf of Aden).

- We deeply thank the referee's comments and the effort hemade in reviewing carefully our work. In the new version of the manuscript we have implemented all the points raised in the review.

My major concern is on the analysis and some of them are listed below.

Comment 1: Please add in the text about the criteria used for removing the spikes, out layer and density inversion.

- The paragraph that explain this part has been modified to better explain the quality control process. As a brief explanation, the quality control has been done in three steps: Firstly, spikes and profiles with density inversions have been removed in all the area studied (Red Sea and outer region). Secondly, those profiles in the Red Sea showing temperatures colder than 20°C below 500 m have been removed. This has been done because no temperature below 20°C has been found in the reference KAUST dataset at any depth. Finally, as a third step, for the rest of the profiles (in the Red Sea and outer Region), those lying outside a range defined by three times the standard deviation are also rejected. .

Comment 2: In Figure 2, why is the left panel the out data are plotted, it would be better to keep only the Red Sea data to cope with the caption of the Figure.

- Thank you for your comment. We have discussed about this but we think useful for the reader to see the large amount of misplaced profiles existing in the CORA dataset to better understand the quality control applied.

Comment 3: Figure 5 shows the distribution of all the available observations for January

in the region and the profiles distribution after applying the K-means algorithm.

- That is correct.

Are these profiles shown in (Figure 5b) the only profiles used in Optimal Interpolation?

- Yes, they are. Prior to run the algorithm to obtain the background fields, we have carried out some tests to know the minimum number of observations required to get the analysis field. As you point out, in Figure 5b there are 135 profiles to obtain the background of January. If we had used more profiles, the computational cost (the inversion of the covariance matrix between observations) would have been higher to obtain basically the same result.

Comment 4: When I check the data availability in the Red Sea region from World Ocean Database, the data points are mostly aligned along the center with significantly lower number profiles towards both eastern and western coast. To what extent the second source of data cover this in space and time?

- Thank you for your appreciation. In fact, we spent some time comparing WOD data and CORA data while preparing the manuscript. In order to clarify this aspect, we attached two figures below(Figure 1-SC1 and Figure 2-SC1). The first one compares the number of observations between both datasets for three different years. The second figure shows the number of observations per year in both datasets. It can be seen that CORA includes more profiles and a better coverage than WOD.

Comment5: The 3D gridded temperature product spanning for the period 1958-2017 is will be very helpful in understanding the Red Sea. From my understanding of the manuscript, I found that the amount of profiles in the Red Sea used for the analysis is very low, except for 2 or 3 years (1959, 2000 and 2016). If this is true, is the derived product will be reliable to discuss interannual and decadal signal?

- We believe the product is reliable to assess the interannual and decadal signal. First, we have to say that the number of observations is not the only thing that matters, as

the spatial distribution of those observations is also very important (i.e. with less than 10 profiles one can obtain a good representation of the large scale patterns if they are well placed). Second, the Optimal Interpolation algorithm also produced an estimate of the error associated to each analysis field depending on the number of observations and their spatial distribution (.ie. the formal error). We deliver that formal error along with the TEMPERSEA product. This can help to identify the periods when the product is less reliable and to quantify those errors.

In the discussion section we show that 10 observations per month in the Red Sea would be enough to do a reliable mapping (see Fig 21 in the paper). Moreover, to reinforce the confidence in the product we compare the results with two source of satellite data and the results are within the error bar (see Fig 10 in the paper).

Comment 6: A table explaining the number of profiles used in the OI per each decade separately in the Red Sea will be helpful to show the data distribution in the Red Sea (which is the prime focus of the study) used in the analysis in addition to a map showing the data spread can be added as supplementary file.

- We think that separating the number of observations per decade would not provide any new information as in Figure 2-SC1 we represent the number of observations per year. In order to clarify your question, see below the Figure 3-SC1 and Table 1 (see in Supplement pdf file) which show the number of observations per decade in the Red Sea and the outer region separately. Nevertheless, we emphasize that what it is important to evaluate the reliability of the product are the number and distribution of the observations per month. So, using the number of profiles per decade would not produce a reliable estimate of the product accuracy. Instead, the most accurate approach to assess the reliability of the product is to use the formal error. It is also included in TEMPERSEA product and will be made freely available at PANGEA repository once the paper for publication.

Comment 7: Most of the data represent the outer region and few only represent the

Red Sea, so the title of the manuscript and the name of the product should consider that. . -We thank the reviewer for the comment. Our main interest is the Red Sea and we use the outer data to put Red Sea variability in context. Nevertheless, we accept the reviewer's suggestion and have modified the title of the paper which now reads:

"Temporal evolution of temperatures in the Red Sea and the Gulf of Aden based on in-situ observations (1958-2017)"

Please also note the supplement to this comment: https://www.ocean-sci-discuss.net/os-2019-66/os-2019-66-SC1-supplement.pdf

Please also note the supplement to this comment:
https://www.ocean-sci-discuss.net/os-2019-66/os-2019-66-AC2-supplement.pdf

———————————————————

[Figure]

**Fig. 1.**

[Figure]

**Fig. 2.**

[Figure]

**Fig. 3.**

---

## Author Comment (AC3) · 5 Nov 2019

Anonymous Referee #2 This study investigates the temperature distribution in the Red Sea from observations collected from 1958 to 2017. The authors combine the data from multiple sources and apply a stringent quality control resulting in a high quality data set which is interpolated to produce a gridded climatology. This allows for an understanding of the Red Sea variability.

[Figure]

- We are grateful to the referee for the constructive comments provided and the in depth reading of the present work. We have followed his/her suggestions, which we believe have helped to improve our manuscript.

As the observational data was collected from CTDs the article could have been greatly improved if the authors had included the analysis of salinity and done the calculations along density isopycnals rather than on depth surfaces.

- Thanks for the comment. We also believe that salinity is important, but there have been several reasons for us to not include its analysis in this work. The number of salinity observations in the basin is significantly smaller than the temperature ones. At the same time, the correlation length scales for salinity are smaller than those of temperature (Llases et al, 2016), so more data would be required to obtain a reliable product. Additionally, including salinity would require specific tests to calibrate the algorithm, and to quantify the uncertainties, which would involve a huge extra effort For all this, we have prefered to focus on temperature characterization, specially considering that temperature has been recognized as the most influential factor for Red Sea ecosystems. We hope in the near future there will be enough salinity profiles thanks to the new observational systems that will allow us to produce an equivalent product for the salinity.

Furthermore the temperature used needs to be either Conservative Temperature or potential temperature not in situ temperature.

- The reviewer is right and in fact potential temperature has been used. By in-situ we aimed at differentiating the in-situ observations from the satellite observations used afterwards. We have included the term "potential temperature" in the first paragraph of section 2.1.

I was surprised by the high percentage of the observations data was located incorrectly, C1

are the authors sure there is not a salinity compensation to this low temperature water that produces an appropriate density for this region.

- Thanks for your appreciation. We had carefully checked that extent prior to discarding those profiles, but we are sure that there is no salinity compensation.

Overall I found the paper to be well written and is interesting and I believe it should be published. It is great that the authors made TEMPERSEA freely available.

- Thank you very much. The product will be made freely available at PANGEA repository once the paper is accepted by the journal.

---

## Author Comment (AC4) · 5 Nov 2019

This paper takes generally available in situ temperature profile data for the Red Sea and Gulf of Aden, combines it with newly available data to create long-term climatological mean fields of surface and subsurface temperature as a baseline for time series of month/year temperature fields (surface and subsurface) for all months for

years 1958-2017. Error estimates are calculated from subsampled GLORYS reanaly-sis data. Some discussion of season, interannual, and decadal variability is included, with decadal trends of opposite sign at the surface and at 125 m depth. This work is definitely of interest, both for the climatological mean fields of tempera- ture in the Red Sea and Gulf of Aden, and for the analysis of seasonal to decadal changes in the temperature field, with their influence on the climate and biota of the region. The authors write very clearly regarding the method used, with a particularly C1

nice explanation of optimal interpolation and of the calculation of error statistics. A more thorough examination of the data would improve the paper, as would validation of the subsurface long-term mean fields against existing products, and more discussion of results, particularly trends of opposite sign at different levels in the water column. Details below.

-We thank the reviewer for his/her overall positive evaluation. In the following we try to address all the his/her comments.

First, the addition of the KAUST data set is a welcome augmentation of existing data for the Red Sea, especially with the possibility of continued monitoring by this source. I do not know the data policy for this journal, but the data used within the paper should be publicly available for reproducibility. The authors should note in the paper where the data can be obtained.

- The final product will be made freely available in the Pangea repository once the paper is accepted. This sentence is added at the end of the paper, in the acknowledgments section.

Figure 2 shows a rather startling distribution of temperature values in the Red Sea, especially with what appears to be a very large number of profiles with temperatures well outside the range of Red Sea temperatures at deeper depths. It would be a great service if the authors could detail the data a little more especially those which they state must have erroneous positions. This would help users (and maintainers) of CORA and

similar data sets to examine and either flag or correct the erroneous data. Did the authors use CORA quality flags? Did these erroneous data have CORA quality flags?

- Yes, we used the CORA quality flags to discard suspicious profiles. Specifically, we downloaded quality flags related to temperature and depth and only kept those profiles flagged with a value of 1 (Good data). It must be said that as a previous control we also tried to keep observations with flags equal to 2 (Probably good data), with the intention of applying a postprocessing, but this approximation does not increased the number of good profiles. Therefore, we decided to use the more restrictive selection keeping only profiles flagged as "Good Data". Regarding the control of erroneous positions, CORA quality control process considers "bad location" those profiles on land positions (positions more than 5km distant from nearest coastline with elevation above 50m). As you can see in the next figure (Figure 1-RC3) for the Red Sea, no observations are located on land, so CORA flags were not available to identify the mislocation of the profiles we have discarded. The full description of CORA database flags are obtained from (http://resources.marine.copernicus.eu/documents/QUID/CMEMS-INS-QUID-013-001b.pdf).

Figure 7shows a patter of RMSE Glorys vs. climatology (and optimal algorithm) that appears suspicious -with what looks like the exact same pattern in the 1960s, 1980s, and 2000s centered at 1000 m with the intermittent decades showing near zero error. Can the authors explain this?Is it some kind of decadal cycle embedded in Glorys, rendering it maybe less than useful for error analysis?

- The reviewer is right noticing this periodicity in the diagnostic. The reason is that, in order to obtain RMSE Glorys Vs Optimal Algorithm (Figure 7b in the paper) we needed to extract the observation locations of observations for the whole period of CORA (60 years, 1958-2017), but Glorys record is only 23 years long (1993-2015). Therefore, we concatenate the 23 years of Glorys till cover the time of observations, so we can extract the CORA locations for the whole period. It must be noted that we do not really care about the actual values of Glorys. We only use it as a synthetic reality so we can

Interactive
comment

test the impact of the mapping procedure.

It also might be nice to enlarge the upper few hundred meters where the largest errors are found, but hard to see in the full vertical graphic.

- We thank the reviewer for the suggestion. We have modified the vertical axis of Figure 7 to increase the zoom in the upper layer.

The long-term climatological mean field is discussed at length, but only validated with a comparison with AVHRR at the surface. It should be compared at subsurface depths to the World Ocean Atlas 2018 (WOA18) field, which are on the same grid size (0.25 x 0.25) and over nearly the same time period (1955-2017) - or another widely used long-term global climatological mean field. This comparison could yield some interesting results as to the efficacy of concentrating on a specific region, instead of using a region of a global climatology, with attendant extra attention, quality control, and in this case new data sources. C2

- Thank you for this suggestion, the comparison with another widely used product is really worth to be done. We have checked the availability of WOA18 database but just climatology fields are available at the repository. Therefore we have compared our TEMPERSEA product with another well-known hydrographic gridded product used in the IPCC reports (Ishii et al., 2003) In spite of having coarser spatial resolution (1°), it provides monthly field temperatures from 1955 to 2012, thus allowing a more in – depth comparison for the common period. Several diagnostics have been computed. First, we compare the annual mean temperature at three different depths (at surface, at 125m and 325 m of depth), both for the Red Sea and the Gulf of Aden (Figure 2-RC3 and Figure 3-RC3). It can be seen that both products are highly correlated in the upper layer, while they differ much more in the subsurface layers. This is also confirmed by the second diagnostic, the spatially averaged RMS difference computed each month for the whole domain (Figure 4-RC3). There is a clear maximum in the RMSD at 125m. Unfortunately, there is not independent data that could be used to
decide which product is more accurate. Therefore, we have decided to not include this comparison in the paper as we are not able to show the added value of the product.

Grid size - sampling strategy: is a 0.25 x 0.25 degree grid really necessary to capture temperature change in the Red Sea? According to the authors discussion, less than 10 temperature profiles per month are necessary to adequately quantify temperature change in the Red Sea. If that is truly the case, would not a 1.0 x 1.0 grid along the axis of the Red Sea be sufficient to capture temperature change?

- To better define the changes of the temperature along the abrupt coast of the Red Sea and his characteristic strait in the South, it is necessary to work with a relatively fine grid. Even if the final structures have large characterisic length scales, we prefer to provide the data in a way that properly capture the coastlines. Moreover, there is another mathematical reason. If the characteristic length scales are about 100-150km (as computed from the Glorys data), we need at least 4 grid points to properly capture those structures, so 0.25° is the minimum resolution to be in the safe side.

In figure 4, it is very hard to see the grid structure used - is there another way to represent it? Maybe just in black and white rather than color?

- Thanks for the suggestion, we have modified the figure in the paper.

But assuming there are multiple grids laterally across the Red Sea at each latitude, it appears that the K-mean algorithm aggregates data into one or sometimes two grid areas across the Sea longitudinally (Figure 5). These appears to lose any advantage of a 0.25 x 0.25 grid resolution. It may be due to the graphic, but the authors should spend some more time discussing the importance of the 0.25 x 0.25 grid resolution to this work.

- We think the reviewer has misinterpreted the figure. The goal of the K-means is to reduce the number of observations at the time of computing the background field. We do that clustering them, so we can remove points that would provide redundant
information for the computation of the climatology. Then, the spatial analysis for the background field is performed on the standard 0.25° grid. For the monthly analysis the number of observations is much reduced so we can use all of them in the mapping procedure and take advantage of the periods/locations when/where there are many observations.

Time frequency: similarly, what is the advantage of the month/year time frequency (12 monthly temperature fields in the Red Sea per year 1958-2017)? As the authors note (with the term "surprisingly" though I dont think it should be surprising to the authors who are familiar with historic measurement strategies in the Red Sea) there are many months without any data at all in the Red Sea, and other months with very few measurements. Seasonal temperature cycle in the Red Sea is examined from a climatological (long-term) perspective. I dont see any particular explanatoin of the advantage to month/year fields over simple yearly fields in quantifying and discussing interannual and decadal variability, especially for data sparse years. The authors should do a little more explanation of why monthly fields are produced. At the least a matrix of coverage (or lack thereof) for each month/year should be presented graphically. This would give a better understanding of data sparsity influence on error, as a companion to figure 21. C3

- Even if in our analysis we only make advantage of the monthly fields when computing monthly std variability, we strongly believe that it is better to provide the product at the highest time resolution. Then, the users could decide at which level would they like to aggregate the data. It has to be kept in mind that during some periods there were enough profiles to accurately characterize monthly variations as can be seen in the following figure (Figure 5-RC3). In the product the periods of better quality are reflected in the error maps and error time series as discussed in the text.

In discussing results, the authors note that most interannual variations in the upper layer of the Red Sea can be explained by large scale changes in the air temperature. This is not completely convincing. There is a good correlation, but isnt it equally as

likely that it is the air temperatures influenced by the upper ocean temperature rather than the other way round? Short wave radiation as well as trapped long wave radiation is absorbed by the ocean surface and radiated back at a slower rate to the lower atmosphere. A little more discussion would be needed to convince that it is large scale air temperature which is the major factor in the upper ocean.

- When preparing the first version of the manuscript we had a thorough discussion with atmosphere scientists about this issue. They suggested to use air temperature at 850mbars (roughly 1500 m height) to ensure that the ocean feedbacks are minimized. We agree that the sea has an effect on the air temperature, but this is restricted to the lower layers. Air temperature at 850 mbars (1500 m height) is too far from being significantly affected by the sea temperature of a small region like the Red Sea. In the manuscript we have added a sentence clarifying that 850mbars correspond to 1500m height. Also we discuss the correlation with air temperature at two heights, close to the sea surface (1000 mbars) and at 850 mbars, showing that correlations are higher close to the sea surface due to the air-sea feedbacks, but that correlations with temperature at 850 mbars is still very high.

One of the remarkable features the authors find is that upper ocean temperatures are increasing (decadally) but lower depths are decreasing. How can this be if the main factor in the temperature change is air temperature, and there is little exchange with any water source outside the Red Sea? It may be that the answer has to do with the interannual change in the depth of the thermocline.

Figure 14 shows thermocline depth seasonal change. Thermocline depth in the south is fairly constant over the year, but changes in the north. If the thermocline were to shallow in February say, cooler water would be higher in the water column and heating would be concentrated closer to the surface, creating the opposite sign trend pattern with depth shown by the authors. This is speculation, but it would be worth a bit more investigation by the authors to validate and maybe explain the change in sign for decadal trend.

- We don't really have an explanation for this discrepancy between the long term evolution of both layers, and it is out of the scope of the paper to run a full analysis on this interesting issue. Regarding the interannual change in the depth of the thermocline we do not understand why that would explain the long term discrepancies between layers, as the thermocline depth is a diagnostic, not a mechanism.

Small things - line 98: what does "delayed mode" mean here? - lines 116, 117, if OSTIA and ICOADS are acronyms, they should be defined. - line 132, "sea-ice concentration" maybe could be removed. GLORYS may assimilate but it is irrelevant in the Red Sea. Thanks for the comments. We have updated the paper with those corrections. it would be nice, in figure 3 to give some indication of the data which came from KAUST as opposed to CORA.

- We have included a dashed line in Figure 3 to indicate the observations coming from KAUST.

line 204, add space between "as" and gamma. - line 206, "pof" should be "of"

- Thanks for identifying the typos, they have been corrected (L213 and L215).

lines 315-317, why would satellite data from the top mm of the water column have a larger variability than in situ data from 2-4 m?

- It is stated in the manuscript that the product is representative of the first 4 m of the water column. Therefore, one can expect that that fraction of the water column is less responsive to changes in the forcing that the first mm of the water column (as it involves more mass). Consequently the variability is somehow damped.

line 412, "imposed to" should be "imposed on" - lines 493-494, lateral advection seems to play an important role..." replace "seems to play" with "plays" if there is actual evidence for this. - line 561, "specially" should be "especially"

- Thanks, these have been corrected.

lines 565-566, "Our results show that multidecadal variations have been important in the past and can bias high the trends from 30-40 years of data." How can multidecadal trends, presumably a cycle, bias high trends? Are the authors referring to multidecadal trends which are not fully represented in 30- 40 years?

- Yes, this is exactly what we mean. Multidecadal variations (not trends) not fully represented by the 30-40 years of data can enhance/reduce the underlying long term trends.

It appears from figure 19 that this could be so in this specific case, but as a generality a partial cycle could bias trends either high or low. Authors should either remove "high" or refer specifically to the Red Sea trend.

- We agree, we have removed the adjective "high"

Please also note the supplement to this comment:
https://www.ocean-sci-discuss.net/os-2019-66/os-2019-66-AC4-supplement.pdf

———————————————————

**Locations of CORA profiles before QC**

Coastline

**Fig. 1.**

[Figure]

**Fig. 2.**

[Figure]

**Fig. 3.**

[Figure]

[Figure]

**Fig. 4.**

**Nº obs Red Sea**

**Nº obs Outer Region**

**Fig. 5.**

---

## Author Comment (AC5) · 5 Nov 2019

Dear Trevor,

We feel that we have fully addressed all discussion contributions. As we address the issues in the interactive discussion we have been updating the manuscript at the same time. In this way, I attach the revised and updated manuscript as pdf with the changes suggested in the discussion phase.

Besides, we believe that you should know that the title of the paper has been updated at the request of SC1, from "Temporal evolution of Red Sea temperatures based on insitu observations (1958–2017)" to the new title "Temporal evolution of temperatures in the Red Sea and the Gulf of Aden based on in-situ observations (1958-2017)".

[Figure]

Best regards,

Miguel Agulles

Please also note the supplement to this comment:
https://www.ocean-sci-discuss.net/os-2019-66/os-2019-66-AC5-supplement.pdf